# Transcriptional correlates of malaria in RTS,S/AS01-vaccinated African children: a matched case–control study

**Gemma Moncunill**[1,2]*, **Jason Carnes**[3§], **William Chad Young**[4§], **Lindsay Carpp**[4§], **Stephen De Rosa**[4], **Joseph J Campo**[5], **Augusto Nhabomba**[6], **Maxmillian Mpina**[7], **Chenjerai Jairoce**[6], **Greg Finak**[4#], **Paige Haas**[3¶], **Carl Muriel**[4], **Phu Van**[4], **Héctor Sanz**[1], **Sheetij Dutta**[8], **Benjamin Mordmüller**[2,9**], **Selidji T Agnandji**[9,10], **Núria Díez-Padrisa**[1], **Nana Aba Williams**[1], **John J Aponte**[1], **Clarissa Valim**[11], **Daniel E Neafsey**[12,13†], **Claudia Daubenberger**[14,15†], **M Juliana McElrath**[4,16‡], **Carlota Dobaño**[1,2‡], **Ken Stuart**[3,4,17,18‡], **Raphael Gottardo**[4,19*‡]

[1]ISGlobal, Hospital Clínic - Universitat de Barcelona, Barcelona, Spain; [2]CIBER de Enfermedades Infecciosas, Madrid, Spain; [3]Center for Global Infectious Disease Research, Seattle Children's Research Institute, Seattle, United States; [4]Vaccine and Infectious Disease Division, Fred Hutchinson Cancer Research Center, Seattle, United States; [5]Antigen Discovery Inc, Irvine, United States; [6]Centro de Investigação em Saúde de Manhiça (CISM), Rua 12, Cambeve, Vila de Manhiça, Maputo, Mozambique; [7]Ifakara Health Institute. Bagamoyo Research and Training Centre, Bagamoyo, United Republic of Tanzania; [8]Walter Reed Army Institute of Research (WRAIR), Silver Spring, United States; [9]Institute of Tropical Medicine and German Center for Infection Research, Tubingen, Germany; [10]Centre de Recherches Médicales de Lambaréné (CERMEL), BP 242, Lambaréné, Gabon; [11]Department of Global Health, Boston University School of Public Health, Boston, United States; [12]Broad Institute of Massachusetts Institute of Technology and Harvard, Boston, United States; [13]Harvard T.H. Chan School of Public Health, Boston, United States; [14]Swiss Tropical and Public Health Institute, Basel, Switzerland; [15]University of Basel, Basel, Switzerland; [16]Departments of Laboratory Medicine and Medicine, University of Washington, Seattle, United States; [17]Department of Pediatrics, University of Washington, Seattle, United States; [18]Department of Global Health, University of Washington, Seattle, United States; [19]University of Lausanne and Centre Hospitalier Universitaire Vaudois, Lausanne, Switzerland

**\*For correspondence:**
gemma.moncunill@isglobal.org (GM);
Raphael.Gottardo@chuv.ch (RG)

[†]These authors contributed equally to this work
[‡]These authors also contributed equally to this work
[§]These authors also contributed equally to this work

**Present address:** [#]Ozette Technologies, Seattle, United States; [¶]Department of Cellular and Molecular Pharmacology, University of California, San Francisco, United States; [**]Department of Medical Microbiology, Radboudumc, Nijmegen, Netherlands

## Abstract

**Background:** In a phase 3 trial in African infants and children, the RTS,S/AS01 vaccine (GSK) showed moderate efficacy against clinical malaria. We sought to further understand RTS,S/AS01-induced immune responses associated with vaccine protection.

**Methods:** Applying the blood transcriptional module (BTM) framework, we characterized the transcriptomic response to RTS,S/AS01 vaccination in antigen-stimulated (and vehicle control) peripheral blood mononuclear cells sampled from a subset of trial participants at baseline and month 3 (1-month post-third dose). Using a matched case–control study design, we evaluated which of these 'RTS,S/AS01 signature BTMs' associated with malaria case status in RTS,S/AS01 vaccinees. Antigen-specific T-cell responses were analyzed by flow cytometry. We also performed a cross-study correlates analysis where we assessed the generalizability of our findings across

three controlled human malaria infection studies of healthy, malaria-naive adult RTS,S/AS01 recipients.

**Results:** RTS,S/AS01 vaccination was associated with downregulation of B-cell and monocyte-related BTMs and upregulation of T-cell-related BTMs, as well as higher month 3 (vs. baseline) circumsporozoite protein-specific CD4+ T-cell responses. There were few RTS,S/AS01-associated BTMs whose month 3 levels correlated with malaria risk. In contrast, baseline levels of BTMs associated with dendritic cells and with monocytes (among others) correlated with malaria risk. The baseline dendritic cell- and monocyte-related BTM correlations with malaria risk appeared to generalize to healthy, malaria-naive adults.

**Conclusions:** A prevaccination transcriptomic signature associates with malaria in RTS,S/AS01-vaccinated African children, and elements of this signature may be broadly generalizable. The consistent presence of monocyte-related modules suggests that certain monocyte subsets may inhibit protective RTS,S/AS01-induced responses.

**Funding:** Funding was obtained from the NIH-NIAID (R01AI095789), NIH-NIAID (U19AI128914), PATH Malaria Vaccine Initiative (MVI), and Ministerio de Economía y Competitividad (Instituto de Salud Carlos III, PI11/00423 and PI14/01422). The RNA-seq project has been funded in whole or in part with Federal funds from the National Institute of Allergy and Infectious Diseases, National Institutes of Health, Department of Health and Human Services, under grant number U19AI110818 to the Broad Institute. This study was also supported by the Vaccine Statistical Support (Bill and Melinda Gates Foundation award INV-008576/OPP1154739 to R.G.). C.D. was the recipient of a Ramon y Cajal Contract from the Ministerio de Economía y Competitividad (RYC-2008-02631). G.M. was the recipient of a Sara Borrell–ISCIII fellowship (CD010/00156) and work was performed with the support of Department of Health, Catalan Government grant (SLT006/17/00109). This research is part of the ISGlobal's Program on the Molecular Mechanisms of Malaria which is partially supported by the Fundación Ramón Areces and we acknowledge support from the Spanish Ministry of Science and Innovation through the 'Centro de Excelencia Severo Ochoa 2019–2023' Program (CEX2018-000806-S), and support from the Generalitat de Catalunya through the CERCA Program.

## Introduction

Malaria remains a serious public health problem, with an estimated 241 million cases and 627,000 related deaths in 2020 (***World Health Organization, 2021a***). Despite the strides that interventions such as long-lasting insecticide-treated bed nets, improved vector control and diagnostic tests, and mass antimalarial drug administration campaigns have made toward reducing malaria-related morbidity and mortality (***Yang et al., 2018***; ***Eisele, 2019***), there is a critical need for an effective malaria vaccine (***Healer et al., 2017***; ***Beeson et al., 2019***).

The RTS,S/AS01 malaria vaccine targets the pre-erythrocytic stage of the parasite life cycle and has been designed to elicit strong humoral and cellular immune responses against the *Plasmodium falciparum* circumsporozoite protein (CSP) (***Hoffman et al., 2015***). This recombinant vaccine consists of a protein containing multiple immunodominant NANP repeats and the carboxy terminus of CSP fused to hepatitis B virus surface antigen (HBs) formulated in the AS01 adjuvant (***Gordon et al., 1995***).

In a phase 3 trial in 15,459 African infants and children (ClinicalTrials.gov NCT00866619) (***Agnandji et al., 2011***; ***RTS,S Clinical Trials Partnership, 2012***; ***RTS,S Clinical Trials Partnership, 2014***; ***RTS,S Clinical Trials Partnership, 2015***), RTS,S/AS01 demonstrated 56% vaccine efficacy (VE) against clinical malaria (follow-up time: 12 month post-last dose) in children aged 5–17 months at enrollment and 31% in infants aged 6–12 weeks at enrollment. In 2015, RTS,S/AS01 became the first malaria vaccine to receive a positive opinion by the European Medicines Agency under Article 58 (***Hawkes, 2015***), and it was recommended by the World Health Organization (WHO) for a malaria vaccine pilot implementation program in Ghana, Malawi, and Kenya that started in 2019 (***World Health Organization, 2019***). Evidence gathered so far from this program led to the recent WHO recommendation for a wider use of this first malaria vaccine in African children at risk (***World Health Organization, 2021b***).

A critical limitation of the RTS,S vaccine is that VE is moderate (lower in infants than children) and wanes substantially within the first 18 months (***RTS,S Clinical Trials Partnership, 2014***). The identification of immune correlates of protection could help guide iterative vaccine improvements and expedite vaccine evaluation. Excellent work has been done on elucidating correlates of RTS,S/AS01-mediated

protection in healthy, malaria-naive adults using the controlled human malaria infection (CHMI) model (*Ockenhouse et al., 2015*; *Chaudhury et al., 2016*; *Kazmin et al., 2017*; *Du et al., 2020*; *Pallikkuth et al., 2020*; *Suscovich et al., 2020*; *Dennison et al., 2021*), and cohort studies in African infants and children have implicated vaccine-induced anti-CSP antibodies (*Dobaño et al., 2019a*; *Dobaño et al., 2019b*; *Ubillos et al., 2018*), as well as CSP-specific Th1 cytokines (*Moncunill et al., 2017b*), and CD4[+] T cells (albeit to a lesser extent) in protection in this population (reviewed in *Moris et al., 2018*).

The MAL067 study, an ancillary study to the RTS,S/AS01 phase 3 trial, was conducted to address key knowledge gaps of RTS,S-induced immune responses and their correlation with protection against natural exposure in the field. Using RNA-sequencing (RNA-seq) data from antigen- or vehicle-stimulated peripheral blood mononuclear cell (PBMC) obtained at baseline and 1-month postfinal primary vaccination dose from infants and children enrolled in Bagamoyo, Tanzania and Manhiça, Mozambique, we aimed to identify baseline and/or RTS,S/AS01-induced signatures associated with clinical malaria risk. Postvaccination anti-CSP antibody levels, cytokine profiles, and T-cell responses, the latter of which were additionally assessed in samples from participants enrolled in Lambaréné, Gabon, were also examined as correlates of clinical malaria and/or of RTS,S/AS01-induced transcriptional responses.

The major finding of our study is that prevaccination expression of immune-related blood transcriptional modules (BTMs), including BTMs related to dendritic cells and monocytes, correlated positively with malaria risk in RTS,S/AS01-vaccinated African children; moreover, the dendritic cell- and monocyte-related elements of this signature appeared to generalize to malaria-naive RTS,S/AS01-vaccinated healthy adults.

# Materials and methods

## Key resources table

| Reagent type (species) or resource | Designation | Source or reference | Identifiers | Additional information |
|---|---|---|---|---|
| Antibody | anti-CD4, clone SK3 (mouse monoclonal) | BD | Cat# 563,550 | 1.5 µl/50 µl staining volume; doi:10.1002/cyto.a.22580 |
| Antibody | anti-CD19, clone SJ25C1 (mouse monoclonal) | BD | Cat# 564,303 | 1 µl/50 µl staining volume; doi:10.1002/cyto.a.22580 |
| Antibody | anti-CD25, clone M-A251 (mouse monoclonal) | BD | Cat# 562,442 | 5 µl/50 µl staining volume; doi:10.1002/cyto.a.22580 |
| Antibody | anti-HLA-DR, clone B169414 (mouse monoclonal) | BioLegend | Cat# 307,637 | 0.625 µl/50 µl staining volume; doi:10.1002/cyto.a.22580 |
| Antibody | anti-CD56, clone HCD56 (mouse monoclonal) | BioLegend | Cat# 318,334 | 0.625 µl/50 µl staining volume; doi:10.1002/cyto.a.22580 |
| Antibody | anti-CD45RA, clone HI100 (mouse monoclonal) | BioLegend | Cat# 304,135 | 0.625 µl/50 µl staining volume; doi:10.1002/cyto.a.22580 |
| Antibody | anti-CD14, clone MφP9 (mouse monoclonal) | BD | Cat# 563,373 | 0.2 µl/50 µl staining volume; doi:10.1002/cyto.a.22580 |
| Antibody | anti-CCR7, clone G043H7 (mouse monoclonal) | BioLegend | Cat# 353,229 | 4 µl/50 µl staining volume; doi:10.1002/cyto.a.22580 |
| Antibody | anti-CD57, clone NK-1 (mouse monoclonal) | BD | Cat# 555,619 | 5 µl/50 µl staining volume; doi:10.1002/cyto.a.22580 |
| Antibody | anti-CD8, clone SK1 (mouse monoclonal) | BD | Cat# 341,051 | 2 µl/50 µl staining volume; doi:10.1002/cyto.a.22580 |
| Antibody | anti-Vδ2 TCR, clone B6 (mouse monoclonal) | BioLegend | Cat# 331,408 | 0.156 µl/50 µl staining volume; doi:10.1002/cyto.a.22580 |
| Antibody | anti-CD3, clone UCHT1 (mouse monoclonal) | Beckman Coulter | Cat# IM2705U | 1 µl/50 µl staining volume; doi:10.1002/cyto.a.22580 |

*Continued on next page*

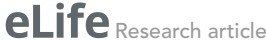

_

*Continued*

| Reagent type (species) or resource | Designation | Source or reference | Identifiers | Additional information |
| --- | --- | --- | --- | --- |
| Antibody | anti-CD38, clone HIT2 (mouse monoclonal) | BD | Cat# 555,461 | 10 µl/50 µl staining volume; doi:10.1002/cyto.a.22580 |
| Antibody | anti-γ/δ TCR, clone 11F2 (mouse monoclonal) | BD | Cat# 655,434 | 1.25 µl/50 µl staining volume; doi:10.1002/cyto.a.22580 |
| Antibody | anti-CD127, clone A019D5 (mouse monoclonal) | BioLegend | Cat# 351,315 | 0.2 µl/50 µl staining volume; doi:10.1002/cyto.a.22580 |
| Antibody | anti-NKG2C, clone 134,591 (mouse monoclonal) | R&D Systems | Cat# FAB138N | 1.25 µl/50 µl staining volume; doi:10.1002/cyto.a.22580 |
| Antibody | anti-CD16, clone 3G8 (mouse monoclonal) | BD | Cat# 557,758 | 0.312 µl/50 µl staining volume; doi:10.1002/cyto.a.22580 |
| Antibody | anti-CD14, clone M5E2 (mouse monoclonal) | BioLegend | Cat# 301,842 | Fluorochrome: BV510 (detected in the same channel as AViD) |
| Antibody | anti-CD56, clone NCAM16.2 (mouse monoclonal) | BD | Cat# 564,447 | Fluorochrome: BUV737 |
| Antibody | anti-CD3, clone UCHT1 (mouse monoclonal) | BioLegend | Cat# 300,436 | Fluorochrome: BV570 |
| Antibody | anti-CD8, clone RPA-T8 (mouse monoclonal) | BD | Cat# 563,821 | Fluorochrome: BV650 |
| Antibody | anti-CD45RA, clone HI100 (mouse monoclonal) | BD | Cat# 560,674 | Fluorochrome: APC-H7 |
| Antibody | anti-CXCR5, clone J252D4 (mouse monoclonal) | BioLegend | Cat# 356,928 | Fluorochrome: PE-Dazzle594 |
| Antibody | anti-PD-1, clone eBioJ105 (mouse monoclonal) | eBioscience | Cat# 25-2799-42 | Fluorochrome: PE-Cy7 |
| Antibody | anti-IFN-γ, clone B27 (mouse monoclonal) | BD | Cat# 560,371 | Fluorochrome: V450 |
| Antibody | anti-IL-2, clone MQ1-17H12 (rat monoclonal) | BD | Cat# 559,334 | Fluorochrome: PE |
| Antibody | anti-IL-4, clone MP4-25D2 (rat monoclonal) | BioLegend | Cat# 500,822 | Fluorochrome: PerCP-Cy5.5 |
| Antibody | anti-IL-13, clone JES10-5A2 (rat monoclonal) | BD | Cat# 564,288 | Fluorochrome: BV711 |
| Antibody | anti-IL-21, clone 3A3-N2 (mouse monoclonal) | Miltenyi Biotec | Cat# 130-120-702 | Fluorochrome: APC |
| Antibody | anti-TNF-α, clone mAb11 (mouse monoclonal) | eBioscience | Cat# 11-7349-82 | Fluorochrome: FITC |
| Antibody | anti-CD40L, clone 24–31 (mouse monoclonal) (mouse monoclonal) | BioLegend | Cat# 310,825 | Fluorochrome: BV605 |
| Antibody | anti-Granzyme B, clone GB11 (mouse monoclonal) | BD | Cat# 560,213 | Fluorochrome: Alx700 |
| Chemical compound, drug | BD FACS Lyse Solution, 10× | BD | Cat #349,202 | doi:10.1002/cyto.a.22590 VC |
| Chemical compound, drug | BD FACS Perm II, 10× | BD | Cat #340,973 | doi:10.1002/cyto.a.22590VC |
| Chemical compound, drug | Brefeldin A | Sigma Chemical Co. | Cat #B-7651 | Final concentration of 10 µg/ml doi:10.1002/cyto.a.22590 VC |
| Chemical compound, drug | CD28/49d (BD Biosciences) | BD | Cat #347,690 | Final concentration of 1 µg/ml doi:10.1002/cyto.a.22590 VC |
| Chemical compound, drug | Golgi Stop containing monensin | BD | Cat #554,724 | doi:10.1002/cyto.a.22590 VC |
| Peptide, recombinant protein | Recombinant AMA1 | WRAIR | | FVO strain, GMP produced in *E. coli* |
| Peptide, recombinant protein | CSP peptide pool | doi: 10.3389/fimmu.2017.01008 Biosynthan (RNA-sequencing stimulations) and Biosynthesis (ICS stimulations) | | |

*Continued on next page*

*Continued*

| Reagent type (species) or resource | Designation | Source or reference | Identifiers | Additional information |
|---|---|---|---|---|
| Peptide, recombinant protein | HBS peptide pool | doi: 10.3389/fimmu.2017.01008 Biosynthan (RNAseq stimulations) and Biosynthesis (ICS stimulations) | | |
| Sequence-based reagent | Universal adapter E5V6NEXT: 5'-iCiGiCACACTCT TTCCCTACACGACGCrGrGrG-3' | Integrated DNA Technologies | | iC: iso-dC, iG: iso-dG, rG: RNA G |
| Sequence-based reagent | Barcoded adapter E3V6NEXT: 5'-/5Biosg/ACACTCTTTCCCT ACACGACGCTCTTCCGATC T[BC6]N10T30VN-3' | Integrated DNA Technologies | | 5Biosg = 5' biotin, [BC6] = 6 bp barcode specific to each cell/well, N10 = unique molecular identifiers, 10 bp |
| Sequence-based reagent | SINGV6 primer: 5'-/5Biosg/ACACTC TTTCCCTACACGACGC-3' | Integrated DNA Technologies | | |
| Sequence-based reagent | P5NEXTPT5 primer: 5'-AATGATACGGCGACC ACCGAGATCTACACT CTTTCCCTACACGAC GCTCTTCC*G*A*T*C*T-3' | Integrated DNA Technologies | | * = phosphorothioate bonds |
| Chemical compound, drug | SEB | Sigma Chemical Co. | Cat #S4881 | |
| Commercial assay or kit | SV96 Total RNA Isolation System | Promega | Cat# Z3500 | |
| Commercial assay or kit | DNA Clean & Concentrator-5 column | Zymo Research | Cat# D4004 | |
| Commercial assay or kit | Advantage 2 Polymerase Mix | Takara Bio | Cat# 639,202 | |
| Commercial assay or kit | dsDNA HS Assay | Life Technologies | Cat# Q32851 | |
| Commercial assay or kit | Nextera XT library preparation kit | Illumina | Cat# FC-131–1096 | |
| Commercial assay or kit | QIAquick Gel Extraction Kit | Qiagen | Cat# 28706 × 4 | |
| Chemical compound, drug | DMSO | Sigma | Cat# D2650 | |
| Software, algorithm | R | The R Foundation | R version 4.0.4 (2021-02-15) | |
| Software, algorithm | Burrows-Wheeler Aligner (BWA) | https://sourceforge.net/projects/bio-bwa/ | BWA Aln version 0.7.10 | |
| Software, algorithm | FlowJo | BD Life Sciences | FlowJo version 9.9 Tree Star | |
| Other | RLT buffer | Qiagen | Cat# 79,216 | |
| Other | RNA protect | Qiagen | Cat# 76,104 | |
| Other | 96-Well V-bottomed plate | Kisker, AttendBio | Cat# G096-VB | |
| Other | Adhesive foil | Kisker, AttendBio | Cat# G071-P | |
| Other | AviD | Invitrogen | Cat# L34957 | 0.5 µl reagent/50 µl staining volume doi:10.1002/cyto.a.22590 VC |
| Other | Maxima H Minus Reverse Transcriptase | Thermo Scientific | Cat# EP0751 | |
| Other | Exonuclease I | New England BioLabs | Cat# M0293S | |
| Other | Agencourt AMPure XP magnetic beads | Beckman Coulter | Cat# A63881 | 0.6× |
| Other | E-Gel EX Gel, 2% | Thermo Fisher | Cat# G401002 | |
| Other | RNA 6000 Pico Chip | Agilent | Cat# 5067-1513 | |

## MAL067 trial

During the MAL055 phase 3 study (ClinicalTrials.gov identifier NCT00866619; *RTS,S Clinical Trials Partnership, 2015*), infants (6–12 weeks) and children (5–17 months) received RTS,S/AS01 or comparator (rabies vaccine for children; meningococcal C conjugate vaccine for infants), with injections given at month 0 (baseline), month 1, and month 2 (*Figure 1*). MAL067 was a multicenter immunology ancillary study nested within MAL055 and selection of participants for MAL067 is described in *Moncunill et al., 2017b*. Of the seven trial sites included in MAL067, three research centers desired and had the required facilities already established to participate in the cellular component of the MAL067 immunology study: Ifakara Health Institute and Bagamoyo Research and Training Centre (IHI-BRTC in Tanzania), Centre de Recherches Médicales de Lambaréné, Albert Schweitzer Hospital (CERMEL, Gabon), and Manhiça Health Research Center, Fundação Manhiça (FM-CISM, Mozambique). PBMC samples were collected at baseline (only children) and again at month 3 (1-month post-third dose) at the three sites. The present study analyzes PBMC data from children in Bagamoyo and from infants and children in Manhiça, as well as from infants and children in Lambaréné (only intracellular cytokine staining [ICS]/immunophenotyping for the latter). All participants met criteria for the modified according-to-protocol (ATP) cohort of the phase 3 trial, from whom we collected PBMC for cellular determinations, and from whom we had available stimulated cells collected. The ATP cohort of the MAL067 immunology study was defined similar to the ATP cohort of the MAL055 clinical trial and is described in detail in *Moncunill et al., 2017b*. In Manhiça all children and infants from one of the recruiting peripheral health posts (Palmeira neighborhood) were included after ethical approvals for the MAL067 immunology study were obtained. After collecting the target sample size of 292 samples in children and infants, we stopped performing fresh stimulation in Manhiça, since per protocol, samples were dedicated to a different study involving B cells. In Bagamoyo, the first 400 children recruited in the phase 3 trial after obtaining ethical approval were included (infants were not). In Lambaréné, after ethical approval, the first 200 volunteers recruited in the phase 3 trial in each age cohort were included. Recruitment period started on 06/08/2009 and ended on 28/01/2011

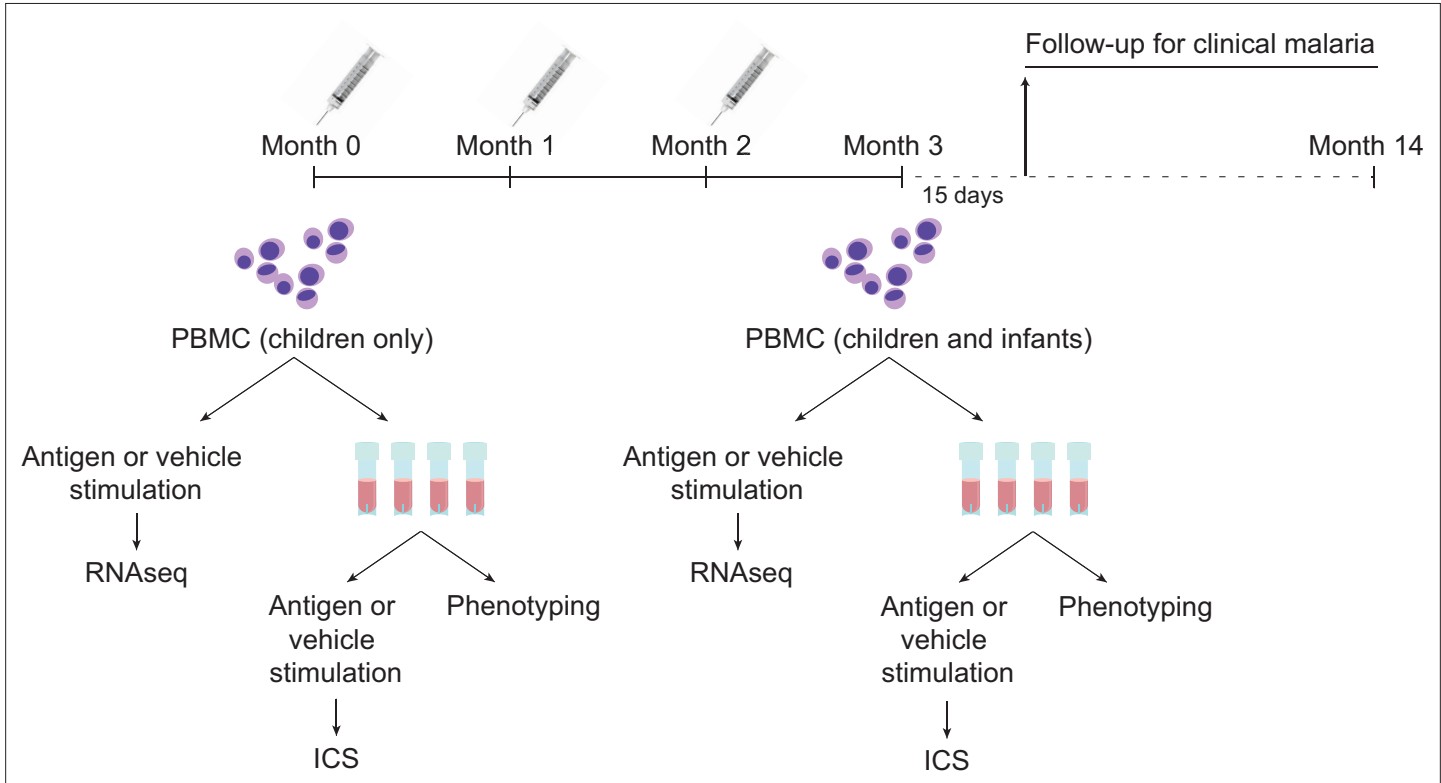

**Figure 1.** Schematic showing vaccination and sampling schedule. Participants received RTS,S/AS01 (or comparator) at months 0, 1, and 2; peripheral blood mononuclear cells (PBMCs) were collected for fresh stimulations and RNA-sequencing and for cryopreservation at months 0 and 3 (1-month postfinal primary vaccination dose). Stim, stimulation.

in Manhiça, 17/09/2009 to 25/02/2010 in Bagamoyo and 22/07/2009 to 31/01/2011 in Lambaréné. PBMC were collected at month 0 before vaccination and approximately 30 days after the third dose of vaccine (month 3) in children and only at month 3 in infants. After cryopreserving $5 \times 10^6$ PBMC in liquid nitrogen, the remaining cells were used for fresh antigen stimulations onsite and the cell pellets were collected. Therefore, only vaccinees with enough PBMC for cryopreservation and additional stimulations (at least $6.6 \times 10^6$ PBMC) were included in this study for RNA-seq.

As malaria transmission intensity at the these three sites was low/moderate (**RTS,S Clinical Trials Partnership, 2015**), we used a case–control design for the study instead of a cohort design. Sample sizes were based on availability of samples and malaria cases. We used all samples available from malaria cases and selected 2–4 matched controls for each case for RTS,S recipients. In selecting controls, we prioritized participants who had samples at both months 0 and 3 and in whom the complete set of antigen stimulations was conducted.

## Case–control definitions

Cases were defined as participants who had any episode of clinical malaria (fever >37.5°C with any parasitemia by blood smear) in the 12 months of follow-up after month 3.5, identified by passive case detection (participants who sought care at a health facility) during all phase 3 trial follow-up. Data were collected during the clinical trial. Controls were participants who did not have any clinical malaria case during the 12 months of follow-up. Controls were matched to cases based on site, age group, and time of vaccination and follow-up. **Supplementary file 1** provides further information on participant characteristics, including select demographics, clinical characteristics, and case–control matching.

## PBMC collection and antigen stimulation

PBMC collection and stimulation with dimethyl sulfoxide (DMSO, vehicle control), apical membrane antigen (AMA1, recombinant protein, FVO strain), CSP (peptide pools), or HBS (peptide pools) is described in **Moncunill et al., 2017b**. Peptides' sequences are detailed in **Moncunill et al., 2017a**. DMSO (Cat# D2650, Sigma) was used at a final dilution of 1/322, the same concentration of DMSO as used for the CSP peptide pool. For stimulation before RNA extraction, $4 \times 10^5$ freshly isolated PBMC seeded in duplicates were rested for 12 hr and then incubated 12 hr at 37°C with 1 µg/ml antigens in 96-well plates. Plates were then centrifuged for 5 min at $250 \times g$ at room temperature and cell pellet duplicates were resuspended and pooled in RLT buffer (Cat# 1053393, Qiagen) at Bagamoyo or RNAprotect (Cat# 76526, Qiagen) at Manhiça, transferred into a 96-well V-bottomed plate (Cat# G096-VB; Kisker, AttendBio) and sealed with adhesive foil (Cat# G071-P; Kisker, Attend Bio) and cryo-preserved at −80°C until RNA extraction.

For stimulation before ICS, cryopreserved PBMC were thawed and then rested in a 37°C, 5% $CO_2$ incubator overnight. PBMC were stimulated for 6 hr with the same peptide pools as above, DMSO (vehicle control), and Staphylococcal enterotoxin B (SEB, Sigma Chemical Co.; Cat #S4881) as a positive control.

## Flow immunophenotyping

Leftover cryopreserved PBMC thawed for ICS (0.5–$1 \times 10^6$ cells) were used for leukocyte phenotyping. The flow cytometry panel and staining protocol used are described in **Moncunill et al., 2014** and in the Key Resources Table. Data were acquired using a BD LSR II flow cytometer (BD Biosciences) directly from 96-well plates using a high throughput sampler. Flow cytometry analysis was performed using FlowJo software (Version 9.9 Tree Star). The gating strategy was performed as in **Moncunill et al., 2014**.

## Intracellular cytokine staining

Antigen- or vehicle-stimulated PBMC were stained using a flow cytometry panel and protocol previously described (**Moncunill et al., 2017a**; **Moncunill et al., 2015**) with the additional marker IL-13. Antibody details can be found in the Key Resources Table. Data were acquired and analyzed as above. Poor quality samples were filtered using two standard criteria: (1) samples with high background (vehicle nonstimulated sample) magnitude was >10% and (2) samples with less than 20,000 CD4 T

cells. No subjects were flagged as high background and 85 were flagged as having low T-cell counts. These were removed from the analysis.

## RNA isolation and sequencing

RNA was extracted at the Center for Global Infectious Disease Research, Seattle Children's Research Institute ( Seattle) using the Promega SV96 Total RNA Isolation kit (Cat# Z3500, Promega) following the manufacturer's protocol. Samples kept in RNAprotect were centrifuged at 4000 × *g* for 7 min at 4°C, cell pellets were resuspended in 150 µl RLT buffer, and 150 µl of 70% ethanol was added prior to processing with the SV96 Total RNA Isolation kit. RNAs were eluted with 100 µl nuclease free water. Each 96-well extraction batch was spot checked by Bioanalyzer using an RNA 6000 Pico chip (Cat# 5067-1513, Agilent) and had an average RIN score of 7.4. RNA samples were distributed in 384-well plates for library preparation. Samples from the same individuals were in the same plate and key study variables (vaccine, site, and cases–controls) were checked for balance across plates to avoid batch effects.

An optimized version of Digital Gene Expression (DGE) was used, based on the Single Cell Barcoding and Sequencing method described by *Soumillon et al., 2014* but further reducing the reverse transcriptase reaction volume. In brief, poly(A)+ mRNA from antigen-stimulated PBMCs was linked to unique molecular identifiers (UMIs) using a template-switching reverse transcriptase (Maxima H Minus Reverse Transcriptase, Cat# EP0751, Thermo Scientific), a universal adapter, and a barcoded adapter (see the Key Resources Table). Then, cDNA from multiple cells was pooled, purified, and concentrated using a DNA Clean & Concentrator-5 column (Cat# D4004, Zymo Research), and treated with Exonuclease I (Cat# M0293S, New England BioLabs). The pooled cDNA was then amplified by single primer PCR using the Advantage 2 Polymerase Mix (Cat# 639202, Takara Bio) and primer SINGV6 (Key Resources Table) and prepped for multiplexed sequencing using a transposon-based fragmentation method (*Adey et al., 2010*), enriching for 3' ends and preserving strand information. Full-length cDNAs were purified with Agencourt AMPure XP magnetic beads (Cat# A63881, 0.6×, Beckman Coulter) and quantified on the Qubit 2.0 Fluorometer (Life Technologies) using the dsDNA HS Assay (Cat# Q32851, Life Technologies). Full-length cDNA was then used with an Nextera XT library preparation kit (Cat# FC-131-1096, Illumina) according to the manufacturer's protocol, except that the i5 primer was replaced by the P5NEXTPT5 primer (see the Key Resources Table). The resulting library was again purified with Agencourt AMPure XP magnetic beads before size selection (300–800 bp) on an E-Gel EX Gel, 2% (Cat# G401002, Thermo Fisher), purification using the QIAquick Gel Extraction Kit (Cat# 28706 × 4, Qiagen) and quantification using the dsDNA HS Assay. Libraries were sequenced at the Broad Institute on Illumina HiSeq paired-end flow cells using an Illumina NextSeq instrument.

## Antibody data analyzed for correlations with BTM expression

NANP-, HBS-, and C-terminal domain of CSP (C-term)-specific antibody data from previous studies were analyzed for correlations with BTM expression as described below. IgG titers (EU/ml) against NANP and against HBS were obtained from the MAL055 trial database (*Agnandji et al., 2011*; *RTS,S Clinical Trials Partnership, 2012*; *RTS,S Clinical Trials Partnership, 2014*; *RTS,S Clinical Trials Partnership, 2015*). IgG concentrations (EU/ml) against NANP and C-term were measured by ELISA at IAVI-HIL (*Dobaño et al., 2019a*). IgG and IgM levels (Median Fluorescence Intensity, MFI) against NANP, C-terminal CSP, and HBS together with 35 RTS,S/AS01 vaccine-unrelated malaria antigens were measured by Luminex technology (*Dobaño et al., 2019b*; *Ubillos et al., 2018*).

## Data processing and statistical analysis

Preprocessing: Preprocessing of RNAseq data was done by Broad Technology Labs. In brief, reads were aligned using BWA Aln version 0.7.10 using UCSC RefSeq (Human 19) with mitochondrial genes added. Quantified samples were then quality controlled using mapping summary statistics to remove low quality samples based on predetermined minimum values for the total number of mapped reads, percent of mapped reads mapped to the human genome, etc. Downstream analysis was applied only to reads that mapped uniquely to a UMI and only mapped to isoforms of the same gene (UMI.unq).

Normalization: The TMM normalization method (*Robinson and Oshlack, 2010*) was applied to account for differing number of read counts and to address unwanted technical variation. The voom

transformation (*Law et al., 2014*) from the limma R package (*Smyth, 2004*) was applied to standardize and appropriately weight the data for use in linear models.

Quality control: In a pilot study, we found that sample libraries that exhibit less than 75,000 total RNAseq reads per sample were of low quality. Thus, such libraries were removed from the study. Genes that had less than 20 samples (around 10%) with read counts greater than 5 were also removed. Multidimensional scaling (MDS) plots as implemented in the plotMDS function of the limma package were used to visualize variability across samples and identify potential sources of variability (batch effects such as total number of reads) or patterns of biological interest (association within experimental factors).

Differential expression: Differential expression was assessed using module-based (using voom and camera [*Wu and Smyth, 2012*]) approaches as implemented in the limma package. Camera, combined with voom, is one of the few gene set enrichment analysis methods that can properly account for intergene correlation in RNA-seq data. Specifically, camera estimates the variance inflation factor for the gene expression that results from intergene correlation in the data and incorporates it into test procedures to control the apparent false discovery rate (FDR). This step is important since significant correlation is expected among genes in the same module. Inference was based on p values adjusted for multiple testing by controlling the FDR with the Benjamini–Hochberg (*Benjamini and Hochberg, 1995*) method. Differential expression was used to downselect modules constituting the PBMC transcriptional response to RTS,S/AS01 vaccination comparing RTS,S/AS01 vaccinees with comparator at month 3 and pre- and post-RTS,S/AS01E vaccination in children (months 3 vs. 0).

BTM analysis: BTMs used were from *Li et al., 2014*. Resulting p values across BTMs (within stimulation condition) were adjusted for multiple testing with a FDR cutoff of 0.2. Only these significant BTMs were tested as candidate immune correlates.

Analysis of antigen (Ag)-specific T-cell transcriptional responses: When analyzing Ag-specific T cells, vehicle-only stimulations (DMSO) were used to determine the effect of Ag stimulation over vehicle stimulation for each PBMC sample. The comparison was performed using the limma package (*Ritchie et al., 2015*) in R as follows: stimulation*vaccine, where stimulation = (HBS, AMA1, CSP) vs. vehicle and vaccine = RTS,S/AS01 vs. Comparator. Quantitative variables were modeled at such, except for age, which was categorized as infant vs. child. Participants with missing data only for certain stimulation were included in the analysis and only the available samples were modeled (no imputation of data was performed).

Equations used were: *Figure 2A*, vehicle: equation = ~plate + total_reads + age + vaccine; CSP, HBS, AMA1: equation = ~plate + total_reads + age + stimulation*vaccine + (1|pid). *Figure 2B*: vehicle: equation = ~plate + total_reads + age + visit + (1|pid); CSP, HBS, AMA1: equation = ~plate + total_reads + age + visit*stimulation + (1|pid), where pid is the patient identifier, modeled as a random effect, and total_reads is the number of sequence reads per sample.

BTM correlations with immunogenicity: For each module, a score was calculated for each RTS,S recipient at months 3 and 0 based on the average normalized expression level of all genes in the modules, on the log scale. Spearman's rank correlation was used to assess association between gene expression, antibody and cellular responses. Each correlation was tested (Spearman correlation test) and a p value was obtained. p values were adjusted within each response (across all gene sets); significance was defined as an adjusted p value ≤0.2.

Correlates analysis: We identified BTMs significantly associated with protection using the limma package. All analyses controlled for plate, total reads, and age (as described above). This model was applied to each BTM and stimulation condition identified in the downselection process. Resulting p values were adjusted for multiple testing with an FDR cutoff of 0.2.

Equations used were: *Figure 3*, vehicle: equation = ~plate + total_reads + age + case; CSP, HBS, AMA1: equation = ~plate + total_reads + age + stimulation*case + (1|pid).

Cross-study correlates analysis: For the cross-study correlates analysis, BTMs were downselected based on month 3 or 0 (as appropriate) data from MAL067 (vehicle-stimulated PBMC only). In brief, month 3 or 0 data for every BTM were tested and FDR adjustment was done across all BTMs. Only those BTMs with FDR < 0.2 in MAL067 were examined as potential correlates of challenge outcome in the CHMI studies, with FDR adjustment performed within each study. The three CHMI studies used in the cross-study immune correlates were: WRAIR 1032 (NCT00075049), which randomly assigned participants to receive RTS,S/AS02A or RTS,S/AS01B at months 0, 1, and 2 (*Kester et al., 2009*)



**Figure 2.** Transcriptional responses and antigen-specific transcriptional responses at 1-month postfinal dose associated with RTS,S/AS01 vaccination. (**A**) Comparison 1: month 3 (M3) peripheral blood mononuclear cells (PBMC), RTS,S/AS01 vs. comparator; (**B**) Comparison 2: M3 PBMC vs. month 0 (M0) PBMC, RTS,S/AS01 recipients only. Cell color intensity represents the significance of the difference in the relevant comparison, expressed as signed $\log_{10}$ false discovery rate (FDR); blood transcriptional modules (BTMs) with significantly different expression (FDR ≤0.2) between the two compared groups are outlined in black. |FDR| < 0.2 (*), <0.05 (**), <0.01 (***). Red, higher expression in RTS,S/AS01 recipients vs. comparator recipients at M3 (Comparison 1) or higher expression in RTS,S/AS01 recipients at M3 vs. M0 (Comparison 2); blue, lower expression in RTS,S/AS01 recipients vs. comparator recipients at M3 (Comparison 1) or lower expression in RTS,S/AS01 recipients at M3 vs. M0 (Comparison 2). High-level BTM annotation groups are shown in the left-most color bar. Numbers of participants in each analysis are: (**A**) Vehicle: 348 (131 comparator, 217 RTS,S/AS01), CSP: 355 (135 comparator, 220 RTS,S/AS01), HBS: 353 (132 comparator, 221 RTS,S/AS01), and AMA1: 351 (132 comparator, 219 RTS,S/AS01). (**B**) Vehicle: 221, CSP: 224 (221 vehicle, 219 CSP), HBS: 225 (221 vehicle, 211 HBS), AMA1: 223 (221 vehicle, 195 AMA1). Numbers include participants not part of the case–control cohort, and thus exceed the numbers in *Table 1*. Each 'vehicle' column displays the vaccine effect in vehicle; each 'stimulation' column displays the vaccine effect for that stimulation compared to vehicle, that is adjusted for vehicle. Detailed equations are given in Methods.

The online version of this article includes the following source data for figure 2:

**Source data 1.** List of blood transcriptional modules (BTMs), p values, and false discovery rates (FDRs) for Comparison 1 (RTS,S/AS01 vs. comparator recipients at month 3).

**Source data 2.** List of blood transcriptional modules (BTMs), p values, and false discovery rates (FDRs) for Comparison 2 (RTS,S/AS01 recipients at months 3 vs. 0).



**Figure 3.** Associations of month 3 levels of RTS,S/AS01 signature blood transcriptional modules (BTMs) with malaria case status in RTS,S/AS01 recipients. Heatmap showing downselected signature BTMs (Comparison 1) with significantly different expression (false discovery rate [FDR] ≤0.2) in month 3 peripheral blood mononuclear cells (PBMC) from RTS,S/AS01 malaria cases vs. nonmalaria controls, in at least one stimulation condition. Cell color intensity represents the significance of the difference in the relevant comparison, expressed as signed $\log_{10}$ FDR; BTMs with significantly different expression in the comparison are outlined in black. |FDR| < 0.2 (*), <0.05 (**), <0.01 (***). Red, higher expression in RTS,S/AS01 cases vs. controls; blue, lower expression in RTS,S/AS01 cases vs. controls. High-level BTM annotation groups are shown in the left-most color bar. Numbers of participants in each analysis are: vehicle: 122, CSP: 123 (122 vehicle, 122 CSP), HBS: 123 (122 vehicle, 115 HBS), AMA1: 123 (122 vehicle, 97 AMA1). The 'vehicle' column displays the vaccine effect in vehicle; each 'stimulation' column displays the vaccine effect for that stimulation compared to vehicle, that is adjusted for vehicle. Detailed equations are given in Methods.

The online version of this article includes the following source data and figure supplement(s) for figure 3:

**Source data 1.** List of blood transcriptional modules (BTMs), p values, and false discovery rates (FDRs) for the comparison of RTS,S/AS01 cases vs. controls at month 3, within each stimulation condition.

**Figure supplement 1.** Associations of month 3 levels of RTS,S/AS01 signature blood transcriptional modules (BTMs) with malaria case status in comparator recipients.

**Figure supplement 1—source data 1.** List of blood transcriptional modules (BTMs), p values, and false discovery rates (FDRs) for the comparison of comparator cases vs. controls at month 3, within each stimulation condition.

MAL068 (NCT01366534), which randomly assigned participants to receive Ad35.CS.01 at month 0 followed by RTS,S/AS01B at months 1 and 2 (heterologous prime–boost) or RTS,S/AS01B at months 0, 1, and 2 (*Ockenhouse et al., 2015*) and MAL071 (NCT01857869), which randomly assigned participants to receive a full dose of RTS,S/AS01B at months 0, 1, and 2 or a full dose of RTS,S/AS01B at

months 0 and 1, followed by a fractional dose at month 7 (*Regules et al., 2016*). Microarray data from WRAIR 1032 were analyzed by *Vahey et al., 2010*, microarray data from MAL068 were analyzed by *Kazmin et al., 2017*, and RNA-seq data from MAL068 and MAL071 were analyzed by *Du et al., 2020*.

Equations used were: MAL067: equation = ~plate + total_reads + age + case; WRAIR 1032, MAL068 RRR, and MAL071 RRR: equation = ~age + infection, MAL067: equation = ~plate + total_reads + age + case; WRAIR 1032, MAL068 RRR, and MAL071 RRR: equation = ~age + infection.

## Results

### Study population and sample collection scheme

PBMC RNA-seq data from a total of 360 participants were analyzed (*Table 1*). For the immunogenicity analysis, 360 participants (225 RTS,S recipients and 135 comparator recipients) were analyzed. For the case–control analysis, baseline RNA-seq data were available for 38 recipients (9 cases and 29 controls) and 19 comparator recipients (5 cases and 14 controls), all of whom were children since baseline samples were not collected from infants. Month 3 RNA-seq data were available for 123 RTS,S/AS01 recipients (31 cases and 92 controls) and 73 comparator recipients (23 cases and 50 controls). All (100%) of the participants in Bagamoyo for whom month 3 RNA-seq data were available were children, whereas nearly all (94.9%) in Manhiça were infants.

*Supplementary file 2* provides similar information for participants for whom immunophenotyping/ICS data were analyzed.

### RTS,S/AS01 vaccination is associated with month 3 downregulation of B-cell- and monocyte-related BTMs, along with upregulation of T-cell-related BTMs

The transcriptional response to RTS,S/AS01 vaccination was assessed in control-stimulated PBMC as well as Ag-stimulated PBMC. Through this approach, we hypothesized that we would see recall responses of Ag-specific T cells activated in vitro, as well as responses of other cell types to the secreted cytokines/chemokines. Of note, the sampling schedule at MAL067 was designed for evaluation of acquired immune responses to the vaccine and not ex vivo responses. Our motivation was that in healthy, malaria-naive adults, the transcriptional response to RTS,S/AS01 has been shown to largely wane by week 3 postfinal dose (*Kazmin et al., 2017*), implying that the majority of the RTS,S/AS01-induced transcriptional changes in this study likely preceded the month 3 sample collection. Three antigens were chosen for stimulation: CSP (peptides covering the CSP region of RTS,S that encodes B- and T-cell epitopes), HBS (peptides covering the HBS, also included in the RTS,S vaccine), and AMA1 (a highly immunogenic antigen expressed briefly on hepatocyte-invading *P. falciparum* sporozoites and predominantly on red blood cell-invading *P. falciparum* merozoites, not present in the RTS,S vaccine; included to analyze naturally acquired immunity responses).

Two comparisons were done to characterize the transcriptional response to RTS,S/AS01 vaccination: Comparison 1: comparing gene expression in month 3 samples from RTS,S/AS01 vs. comparator recipients (month 3 RTS,S/AS01 vs. comparator); and Comparison 2: comparing gene expression in months 3 vs. 0 from RTS,S/AS01 recipients (RTS,S/AS01 months 3 vs. 0). Each comparison has its own advantages: Comparison 1allows the identification of RTS,S/AS01-specific responses while taking into account other environmental factors to which the children are exposed, such as malaria exposure (albeit malaria transmission intensity was low during the study at both sites [*RTS,S Clinical Trials Partnership, 2015*]). Moreover, the very young ages of the trial participants mean that RTS,S/AS01-induced changes may be confounded with normal developmental changes in participant immune systems, further underscoring the value of Comparison 1, as it does not involve comparison across two different timepoints. On the other side, an advantage of Comparison 2is that it takes into consideration each participant's intrinsic baseline gene expression. Comparison 1uses data from both infants and children, whereas Comparison 2can only yield insight into RTS,S/AS01 responses in children (as baseline samples were not collected from infants).

A BTM-based approach was taken to reduce dimensionality, avoid paying a high penalty for multiple testing, and aid results interpretability. For Comparison 1, there were 68 significantly differentially expressed (FDR cutoff ≤0.2) BTMs across all antigen stimulation conditions (*Figure 2*, *Figure 2—source data 1*). The majority (53)of these BTMs were in vehicle-treated PBMCs, with the most

**Table 1.** Numbers, age group, and case–control status of participants by site for whom peripheral blood mononuclear cells (PBMC) RNA-seq data were available at months 0 and/or 3.

**Month 0**

| | Cases (n = 14) | | | | Controls (n = 43) | | | | Not included in the case–control (n = 70) | | | |
|---|---|---|---|---|---|---|---|---|---|---|---|---|
| | Bagomoyo | | Manhiça | | Bagomoyo | | Manhiça | | Bagomoyo | | Manhiça | |
| | Infants | Children | Infants | Children | Infants | Children | Infants | Children | Infants | Children | Infants | Children |
| RTS,S/AS01 (n = 88) | 0 | 9 | 0 | 0 | 0 | 29 | 0 | 0 | 0 | 41 | 0 | 9 |
| Comparator (n = 39) | 0 | 5 | 0 | 0 | 0 | 12 | 0 | 2 | 0 | 19 | 0 | 1 |

**Month 3**

| | Cases (n = 54) | | | | Controls (n = 142) | | | | Not included in the case–control (n = 161) | | | |
|---|---|---|---|---|---|---|---|---|---|---|---|---|
| | Bagomoyo | | Manhiça | | Bagomoyo | | Manhiça | | Bagomoyo | | Manhiça | |
| | Infants | Children | Infants | Children | Infants | Children | Infants | Children | Infants | Children | Infants | Children |
| RTS,S/AS01 (n = 222) | 0 | 19 | 12 | 0 | 0 | 51 | 41 | 0 | 0 | 56 | 28 | 15 |
| Comparator (n = 135) | 0 | 16 | 6 | 1 | 0 | 30 | 16 | 4 | 0 | 29 | 29 | 4 |

common categories being B cells (6 BTMs) and T cells (11 BTMs). Counter to our initial expectations, no significant correlations were identified in the CSP-stimulated cells adjusted by vehicle stimulation. This result is potentially explained by the low frequency of CSP-specific T cells in RTS,S/AS01 vaccinees (e.g., on average, <0.10% of all CD4+ T cells [*Moncunill et al., 2017a*]). In AMA1-stimulated cells adjusted by vehicle stimulation, some correlate BTMs associated with RTS,S/AS01 vaccination were shared with vehicle-treated cells (related to e.g. mitochondria, transcription, and translation), while distinct correlate BTMs were also identified (related to e.g. the cell cycle, dendritic cells, and the nuclear pore). It is possible that the latter finding reflects differences in Ag-specific responses vs. nonspecific responses in vehicle, as the AMA1-stimulated PBMC analysis was adjusted by vehicle stimulation. However, we favor the hypothesis that cytokines/chemokines released from activated T cells in malaria-exposed children and their effects on other PBMC may underlie this difference. Alternatively, AMA1 may be eliciting an innate response (*Bueno et al., 2008*).

For Comparison 2, only RTS,S/AS01 recipients for whom months 0 and 3 samples were available were included in the analysis (i.e., children only). There were a larger number (131) of significantly differentially expressed (FDR cutoff ≤0.2) BTMs across all stimulation conditions; again, the majority (90) were found in vehicle-stimulated PBMC (*Figure 2*, *Figure 2—source data 2*). In vehicle-stimulated PBMC, antiviral/interferon (IFN)-related BTMs were consistently upregulated at months 3 vs. 0, while monocyte- and antigen presentation-related BTMs were consistently downregulated. Similar to Comparison 1, the antigen stimulation results adjusted by vehicle stimulation shared little overlap with the vehicle stimulation results.

## Monocyte-related RTS,S/AS01 signature BTMs associate with clinical malaria risk

To preserve statistical power in the immune correlates analysis, only BTMs differentially expressed after RTS,S/AS01 vaccination according to Comparison 1 (any stimulation) were down selected. We define these 68 BTMs as the 'RTS,S/AS01 signature BTMs' (*Supplementary file 3*). We next investigated if any of the RTS,S/AS01 signature BTMs were associated with clinical malaria case status in RTS,S/AS01 recipients, by comparing expression of the signature BTMs in cases vs. controls, within each stimulation condition.

In vehicle-stimulated PBMC, seven BTMs were significantly differently expressed in RTS,S/AS01 cases vs. controls (*Figure 3*, *Figure 3—source data 1*). Three were associated with risk ('Enriched in myeloid cells and monocytes [M81]', 'Enriched in monocytes (II) [M11.0]', and 'Myeloid cell enriched receptors and transporters [M4.3]'), while four were associated with protection ('Respiratory electron transport chain [mitochondrion] [M219]', 'Respiratory electron transport chain [mitochondrion] [M238]', 'spliceosome [M250]', and 'mitosis [TF motif CCAATNNSNNNGCG] [M169]'). The association of monocyte-related BTMs with risk is consistent with studies reporting a positive correlation between monocyte/lymphocyte ratio and clinical malaria risk and/or severity (*Antwi-Baffour et al., 2018*; *Warimwe et al., 2013b*).

The antigen-specific transcriptional modules associated with clinical malaria risk differed from those seen in vehicle-stimulated PBMC. In CSP- and HBS-stimulated cells adjusted by vehicle stimulation, there were 0 and 1 correlate BTMs, respectively. In AMA1-stimulated cells adjusted by vehicle stimulation, distinct and opposite correlations were seen, for example correlation with protection for 'enriched in activated dendritic cells/monocytes (M64)', 'myeloid cell enriched receptors and transporters (M4.3)', and 'enriched in monocytes (II) (M11.0)'.

An analogous analysis, using the same downselected BTMs, was performed on comparator recipients. For all seven BTMs whose levels in vehicle-stimulated PBMC associated either directly or inversely with risk in RTS,S/AS01 recipients (*Figure 3*), significant correlations were observed in the opposite direction in comparator recipients (*Figure 3—figure supplement 1*, *Figure 3—figure supplement 1—source data 1*), suggesting that positive correlations of the three monocyte-related BTMs with risk and inverse correlations of the mitochondria-related BTMs with risk are specific to RTS,S/AS01 recipients.

**Figure 4.** RTS,S/AS01 vaccination elicits circumsporozoite protein (CSP)-specific polyfunctional T-cell responses that do not correlate with clinical malaria risk. Boxplots show (**A**) polyfunctionality score and (**B**) magnitude (% CD4+ T cells expressing IL2 or TNF-α or CD154) of CSP-specific CD4+ T-cell responses in RTS,S/AS01 recipients as assessed by intracellular cytokine staining of peripheral blood mononuclear cells (PBMC) collected at month 0 (M0) or at month 3 (M3). Each dot represents a single participant. Data plotted include all available months 0 and 3 samples, that is paired months 0–3 samples were not required for plotting. (**C**) Polyfunctionality score and (**D**) magnitude of CSP-specific CD4+ T-cell responses in RTS,S/AS01 vaccine recipients at month 3, stratified by case–control status. In panels A and B, p values were obtained using a mixed-effects model with participant as a random effect. In panels C and D, p values were obtained using a mixed-effects model with match_id as a random effect. Number of participants in each panel is: (**A**) 213 (73 M0 and 182 M3), (**B**) 194 (61 M0, 175 M3), (**C**) 37 cases and 145 controls, and (**D**) 36 cases and 139 controls.

## RTS,S/AS01 vaccination elicits polyfunctional CSP-specific CD4+ T-cell responses that do not correlate with malaria risk

In addition to transcriptional changes, our group has shown previously that RTS,S/AS01 vaccination elicits vaccine-specific antibody and cellular responses in African infants and children (e.g., *Dobaño et al., 2019a*; *Ubillos et al., 2018*; *Moncunill et al., 2017a*). The polyfunctionality score is a summary measure that encapsulates a participant's entire Ag-specific T-cell response after vaccination (*Lin et al., 2015*). Using data from a pilot study of 179 children (none of whom was a malaria case) at the Manhiça and Bagamoyo sites, Moncunill et al. previously showed that MAL067 RTS,S/AS01 recipients have higher month 3 CSP-specific and HBS-specific CD4+ T-cell polyfunctionality scores than comparator recipients (*Moncunill et al., 2017a*). Consistent with this finding, we report that average CSP-specific

CD4[+] T-cell polyfunctionality score is higher at month 3 vs. baseline in RTS,S/AS01 vaccine recipients (*Figure 4A*). The few high responders at baseline can likely be attributed to prior malaria exposure. However, there was no difference in magnitude (frequency of CD4[+] T-cell expressing IL-2 or TNF or CD154) at month 3 vs. baseline in RTS,S/AS01 vaccine recipients (*Figure 4B*), nor was there a difference in average month 3 CSP-specific T-cell response polyfunctionality or magnitude between RTS,S/AS01 cases vs. controls (*Figure 4C, D*).

## Month 3 levels of RTS,S/AS01 signature BTMs tend to correlate directly with month 3 IgM antibody responses and inversely with month 3 IgG responses

We next investigated whether month 3 levels of the RTS,S/AS01 signature BTMs were associated with month 3 humoral immune responses in RTS,S/AS01 vaccinees. In vehicle-treated PBMC, both positive and negative associations were seen for multiple antibody variables across functional categories (*Figure 5*, *Figure 5—source data 1*) but mainly against nonvaccine antigens. Month 3 IgM antibodies against LSA1, MSP1 Block 2 (MAD20 strain), and MSP6 tended to correlate with month 3 levels of DC-, inflammatory/TLR/chemokine-, and monocyte-related BTMs (among others). In contrast, month 3 IgG antibodies against AMA1 (strains 3D7 and FVO) tended to correlate inversely with month 3 levels of DC- and monocyte-related BTMs, among others. These associations were not seen in comparator recipients (*Figure 5—figure supplement 1*), suggesting specificity to RTS,S/AS01 receipt, although we note that sample size is smaller which would have reduced statistical power to detect differences. Month 3 levels of cellular variables assessed by polychromatic flow cytometry did not correlate significantly with the month 3 level of any BTM.

## Cross-study immune correlates analysis reveals a mostly consistent association in RTS,S/AS01 vaccinees between baseline expression of DC- and monocyte-related BTMs and risk

An important question is whether the results of our analysis of the MAL067 trial, which was conducted in African infants and children in malaria-endemic areas, are generally translatable to other study populations. PBMC transcriptomic data are available for at least three different CHMI studies conducted in healthy, malaria-naïve adults in the United States. We performed a cross-study immune correlates analysis where we examined whether the BTMs associated with clinical malaria risk in MAL067 showed similar associations with challenge outcome in each of the three CHMI studies described in Methods: WRAIR 1032, MAL068, and MAL071. Importantly, all these trials share a common vaccine arm: one full dose of RTS,S/AS01B at months 0, 1, and 2 (referred to as the 'RRR' arm). Due to differences in sampling schedules, and the presence of the CHMI challenge (which would complicate results interpretation), we could not compare the exact same month 3 timepoint across studies. We chose instead to compare 21 days post-third dose in MAL068 and in MAL071, that is of day of challenge, and 14 days post-third dose in WRAIR 1032, that is just before or on day of challenge. We refer to these slightly different postvaccination timepoints as 'month 3' for simplicity. The month 3 cross-study correlates analysis included BTMs whose month 3 levels (in vehicle-stimulated PBMC) associated with clinical malaria risk in MAL067 RTS,S/AS01E recipients (*Figure 3*, *Figure 3—source data 1*) and is shown in *Figure 6A*. No BTM was consistently associated with malaria risk (or nonprotection) across all four studies. The most consistent result was for the monocyte-related BTM 'enriched in monocytes (II) (M11.0)', whose month 3 expression was significantly associated with risk in two of the three CHMI studies (*Figure 6A*, *Figure 6—source data 1*).

We next performed the baseline correlates analysis of MAL067 (left-most column, *Figure 6B*). Compared to the results from the month 3 analysis (7 BTMs), the baseline correlates analysis of MAL067 revealed a larger number (45) of BTMs, spanning many functional categories, whose month 0 levels in vehicle-stimulated PBMC nearly all associated with clinical malaria risk in RTS,S/AS01 recipients (*Figure 6B*, *Figure 6—source data 2*). The BTM with the most significant association with risk was 'enriched in monocytes (II) (M11.0)' (FDR = 1.80E−14), followed by 'inflammatory response (M33)' (FDR = 2.45E−07) and 'resting dendritic cell surface signature (S10)' (FDR = 6.03E−07). Only one BTM, 'cell cycle and transcription (M4.0)', was significantly associated with risk across all four studies. Of the 335 genes in this module, 130 were also present in 1 or more of the 6 'monocyte-related' BTMs shown

**Figure 5.** Correlations of month 3 transcriptional and adaptive responses in RTS,S/AS01 vaccine recipients. Heatmap showing correlations between month 3 levels of RTS,S/AS01 signature blood transcriptional modules (BTMs) in vehicle-treated peripheral blood mononuclear cells (PBMC) and month 3 antibody responses. Cell color intensity represents the strength of the correlation; BTM/response pairs with significant correlations (false

*Figure 5 continued on next page*

*Figure 5 continued*

discovery rate [FDR] ≤0.2) are outlined in black. Cell color represents correlation direction: red, positive correlation; blue, negative correlation. High-level BTM annotation groups are shown in the left-most color bar. Number of participants: 30–42.

The online version of this article includes the following source data and figure supplement(s) for figure 5:

**Source data 1.** List of blood transcriptional modules (BTMs) whose month 3 levels in vehicle-treated peripheral blood mononuclear cells (PBMC) correlated significantly with at least one month 3 adaptive response variable in RTS,S/AS01 vaccinees, along with variable details, p value, and false discovery rate (FDR) results.

**Figure supplement 1.** Correlations of transcriptional and adaptive responses in comparator vaccine recipients.

**Figure supplement 1—source data 1.** List of blood transcriptional modules (BTMs) whose month 3 levels in vehicle-treated peripheral blood mononuclear cells (PBMC) correlated significantly with at least one month 3 adaptive response variable in comparator recipients, along with variable details, p value, and false discovery rate (FDR) results.

in *Figure 6* (297 genes total across all 6 BTMs), suggesting that the 'cell cycle' and 'monocyte' results may be picking up the same signal.

Comparing across studies, a fair degree of overlap was seen between the MAL067 associations and the CHMI associations. MAL067 and WRAIR 1032 shared the most BTMs significantly associated with risk (29 BTMs); of these, 12 were also associated with risk in MAL068 RRR. BTMs related to dendritic cells and to monocytes were most consistently associated with risk across these three studies ('resting dendritic cell surface signature [S10]', 'DC surface signature [S5]', 'enriched in dendritic cells [M168]', 'enriched in monocytes [I] [M4.15]', 'enriched in monocytes [II] [M11.0]', 'enriched in monocytes [IV] [M118.0]', and 'monocyte surface signature [S4]', significantly correlated with risk in all three studies).

To gain insight into specific module-member genes that may be involved in the RTS,S/AS01 baseline risk signature, we performed the same analysis on the gene level, that is examined associations with clinical malaria risk for each of the constituent genes in the 45 BTMs shown in *Figure 6*. *Figure 6—figure supplements 1–8* show the gene-level association results within the eight BTMs that were significantly associated with clinical malaria risk in MAL067 and at least two of the three CHMI studies, and had at least one gene in MAL067 that was significantly associated with risk (these eight correspond to M4.0, S10, S5, M168, M4.3, M11.0, M4.15, and S4). Within MAL067, 35 unique genes were shown to significantly associate with malaria risk (*Supplementary file 4*); 9 of these genes (*CCNF*, *MKI67*, *KIF18A*, *NPL*, *RBM47*, *CFD*, *MAFB*, *IL13RA1*, and *CCR1*) also had significant association with nonprotection in one of the CHMI studies. Although no individual gene was significantly associated with risk across >two studies, many showed consistent effect (direction and magnitude) across three studies. This further supports our choice to focus on modules instead of individual genes as GSEA increases power to detect more subtle but coordinated changes in gene expression data that would be missed otherwise. For this same reason, GSEA has been shown to enhance cross-study comparisons (*Subramanian et al., 2005*).

Baseline transcriptional associations with month 3 adaptive responses are presented in *Figure 6—figure supplement 9*. The baseline expression of each of 52 BTMs, spanning a range of functional categories, was significantly and positively correlated with CSP-specific CD4+ T-cell polyfunctionality score and the baseline expression of each of 17 BTMs was also significantly and positively correlated with HBS-specific CD8+ T-cell polyfunctionality score. No significant associations were seen with any antibody responses.

The finding that monocyte-related BTMs were expressed significantly higher in RTS,S/AS01 cases vs. controls at month 3 in three of the four studies examined (*Figure 6A*) and at month 0 in three of the four studies examined (*Figure 6B*) suggested that circulating monocyte frequencies may be higher in cases vs. controls at these two timepoints. To investigate this hypothesis, PBMC from RTS,S/AS01 recipients were assessed by immunophenotyping and flow cytometry. As shown in *Figure 6—figure supplement 10A*, the analysis revealed no significant difference in monocyte frequency in cases vs. controls, at either month 3 or 0. At month 3, both the inflammatory monocyte frequency and inflammatory monocyte/lymphocyte ratio tended to be higher in cases than in controls; however, these differences were not significant (*Figure 6—figure supplement 10B, C*). Thus, these findings do not support that the upregulation of monocyte-related genes in PBMC from cases (vs. controls) is due to higher frequencies of circulating monocytes. A potential explanation for why we identified monocyte-related

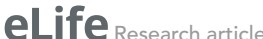

**Figure 6.** Associations of (**A**) month 3 or (**B**) month 0 levels of downselected blood transcriptional modules (BTMs) with malaria case status RTS,S/AS01 vaccine recipients across studies sharing a common months 0, 1, and 2 RTS,S/AS01 arm. (**A**) Heatmap showing the difference in month 3 peripheral blood mononuclear cell (PBMC) BTM expression between RTS,S/AS01 cases vs. controls, in each of three controlled human malaria infection (CHMI) studies, of the seven BTMs whose month 3 levels in vehicle-stimulated PBMC associated with malaria case status in MAL067 (**Figure 3**). 'Month 3' =

*Figure 6 continued on next page*

*Figure 6 continued*

21-day postfinal dose in MAL068 and MAL071, and 14-day postfinal dose in WRAIR 1032. BTMs with significantly different expression (false discovery rate [FDR] ≤0.2, with adjustment done across the five BTMs) are outlined in black. |FDR| < 0.2 (*), <0.05 (**), <0.01 (***). (B) Heatmap showing the 45 BTMs whose month 0 levels showed significantly different expression in MAL067 RTS,S/AS01 malaria cases vs. nonmalaria controls. These 45 BTMs were also examined as potential correlates of challenge outcome in each of the 3 CHMI studies. Significantly different expression is defined as FDR ≤0.2, with adjustment across the 45 BTMs. All data shown are from participants who received the same vaccine regimen: a dose of RTS,S/AS01 at months 0, 1, and 2. Cell color intensity represents the significance of the difference in the case vs. control comparison, expressed as signed $\log_{10}$ FDR; BTMs with significantly different expression (FDR ≤0.2) between the two compared groups are outlined in black. |FDR| < 0.2 (*), <0.05 (**), <0.01 (***). Red, higher expression in RTS,S/AS01 cases vs. controls; blue, lower expression in RTS,S/AS01 cases vs. controls. High-level BTM annotation groups are shown in the left-most color bar. Numbers of participants in each analysis are: (A) MAL067, 122; WRAIR 1032, 39; MAL068 RRR, 21; MAL071 RRR, 16. (B) MAL067, 37; WRAIR 1032, 39; MAL068 RRR, 21; MAL071 RRR, 16. Detailed equations are given in Methods.

The online version of this article includes the following source data and figure supplement(s) for figure 6:

**Source data 1.** List of the seven blood transcriptional modules (BTMs) whose month 3 levels had significantly different expression in RTS,S/AS01 cases vs. controls in MAL067, along with p values and false discovery rate (FDR) results when testing these seven BTMs for significantly different expression in cases vs. controls in the WRAIR 1032, MAL068 RRR, and MAL071 RRR studies.

**Source data 2.** List of the 45 blood transcriptional modules (BTMs) whose month 0 levels had significantly different expression in RTS,S/AS01 cases vs. controls in MAL067, along with p values and false discovery rate (FDR) results when testing these 45 BTMs for significantly different expression in cases vs. controls in the WRAIR 1032, MAL068 RRR, and MAL071 RRR studies.

**Figure supplement 1.** Association of month 0 level of each gene in the 'cell cycle and transcription (M4.0)' module with malaria case status across studies sharing a common months 0, 1, and 2 RTS,S/AS01 arm.

**Figure supplement 2.** Association of month 0 level of each gene in the 'Resting dendritic cell surface signature (S10)' module with malaria case status across studies sharing a common months 0, 1, and 2 RTS,S/AS01 arm.

**Figure supplement 3.** Association of month 0 level of each gene in the 'DC surface signature (S5)' module with malaria case status across studies sharing a common months 0, 1, and 2 RTS,S/AS01 arm.

**Figure supplement 4.** Association of month 0 level of each gene in the 'Enriched in dendritic cells (M168)' module with malaria case status across studies sharing a common months 0, 1, and 2 RTS,S/AS01 arm.

**Figure supplement 5.** Association of month 0 level of each gene in the 'Myeloid cell enriched receptors and transporters (M4.3)' module with malaria case status across studies sharing a common months 0, 1, and 2 RTS,S/AS01 arm.

**Figure supplement 6.** Association of month 0 level of each gene in the 'Enriched in monocytes (II) (M11.0)' module with malaria case status across studies sharing a common months 0, 1, and 2 RTS,S/AS01 arm.

**Figure supplement 7.** Association of month 0 level of each gene in the 'Enriched in monocytes (I) (M4.15)' module with malaria case status across studies sharing a common months 0, 1, and 2 RTS,S/AS01 arm.

**Figure supplement 8.** Association of month 0 level of each gene in the 'Monocyte surface signature (S4)' module with malaria case status across studies sharing a common months 0, 1, and 2 RTS,S/AS01 arm.

**Figure supplement 9.** Correlations of month 0 blood transcriptional module (BTM) expression in vehicle-treated peripheral blood mononuclear cells (PBMC) with month 3 T-cell responses in RTS,S/AS01 vaccine recipients.

**Figure supplement 9—source data 1.** List of blood transcriptional modules (BTMs) whose month 0 levels correlated significantly with at least one month 3 adaptive response variable, along with stimulation, variable details, p value, and false discovery rate (FDR) results.

**Figure supplement 10.** No significant differences in monocyte frequencies in cases vs. controls in RTS,S/AS01 vaccinees at either month 0 or 3.

BTMs in our transcriptional signature of risk yet did not see an association of baseline monocyte frequency, inflammatory frequency, or inflammatory monocyte/lymphocyte ratio with risk is that the baseline monocyte transcriptional signature of risk reflects expression changes in the existing circulating monocyte population, rather than an expansion in the circulating monocyte population.

## Discussion

Our main finding is the identification of a baseline BTM signature that associates with clinical malaria risk in RTS,S/AS01-vaccinated African children. In a cross-study comparison, much of this baseline risk signature – specifically, dendritic cell- and monocyte-related BTMs – was also recapitulated in two of the three CHMI studies in healthy, malaria-naive adults. Our finding fits into a growing body of evidence that baseline immune status can influence vaccine responses (*Tsang et al., 2020*). Fourati et al. showed that higher baseline inflammation (as assessed by transcriptomic profiling and flow cytometric analysis of immune cell subset frequencies) was associated with poor antibody response to the



hepatitis B vaccine (*Fourati et al., 2016*). Tsang et al. showed that baseline interferon signaling was robustly correlated with maximum fold change (postinfluenza vaccination to baseline) in influenza-specific antibody titer (*Tsang et al., 2014*). The HIPC Consortium identified a baseline inflammatory gene signature that was associated with higher antibody responses to influenza vaccine in younger participants, yet lower antibody responses in older participants (*HIPC-CHI Signatures Project Team and HIPC-I Consortium, 2017*). Kotliarov et al. identified baseline transcriptional signatures that predicted antibody responses to the live attenuated yellow fever vaccine and to the trivalent inactivated influenza vaccine (*Kotliarov et al., 2020*). Moreover, Hill et al. reported that increased baseline frequencies of plasmablasts and of circulating T follicular helper cells were associated with higher post-RTS,S/AS01 vaccination antibody titers (*Hill et al., 2020*).

Two previous studies have reported a positive correlation between monocyte to lymphocyte (ML) ratio and clinical malaria risk and/or severity (*Antwi-Baffour et al., 2018*; *Warimwe et al., 2013b*), and a higher ML ratio has been reported to associate with lower VE of RTS,S (*Warimwe et al., 2013a*). Note that the ML ratio in these studies was based on lymphocyte count and monocyte count from a differential blood count performed using a Coulter Counter, and thus cannot inform on the gene expression profiles of the circulating monocytes, or on composition of circulating monocyte subsets. Based on previous evidence in various mouse models of viral infection that inflammatory monocytes inhibit T-cell proliferation (*Norris et al., 2013*; *Mitchell et al., 2012*), T-cell activation (*Mitchell et al., 2012*), and B-cell responses (*Sammicheli et al., 2016*), Warimwe et al. proposed that inflammatory monocytes may inhibit RTS,S-induced protective adaptive responses. Our gene-level correlates analyses suggest an alternative hypothesis, however. With the caveat that the gene-level analyses were performed post hoc, high baseline expression of *STAB1* (which is present in DC-, monocyte-, and cell cycle-related modules) was found to positively associate with malaria risk (*Figure 6—figure supplements 1*, *2* and *6*). *STAB1* encodes stabilin-1 (also called Clever-1), a transmembrane glycoprotein scavenger receptor that links extracellular signals to intracellular vesicle trafficking pathways (*Kzhyshkowska et al., 2006*). Stabilin-1^high monocytes show downregulation of proinflammatory genes, and T cells cocultured with stabilin-1^high monocytes showed decreased antigen recall, suggesting that monocyte stabilin-1 suppresses T-cell activation (*Palani et al., 2016*). Thus, one possibility is that stabilin-1^high immunosuppressive monocytes circulating at baseline could decrease protective RTS,S-induced T-cell responses, or inhibit another aspect of adaptive immunity. Single-cell transcriptomic profiling of PBMC or purified monocyte subsets in future RTS,S trials in African children in malaria-endemic areas could help test this hypothesis.

The RTS,S/AS01 vaccine is adjuvanted with AS01, a liposome-based adjuvant containing 3-*O*-desacyl-monophosphoryl lipid A (MPL) and the saponin QS-21 (*Didierlaurent et al., 2017*). MPL activates the innate immune response by stimulation of Toll-like receptor 4 (TLR4) (*Baldridge et al., 2004*); thus, another interesting finding of the gene-level analyses is the significant association of baseline *TLR4* expression with risk (*Figure 6—figure supplements 1*, *6* and *8*). As *TLR4* is expressed predominantly on monocytes (*Vaure and Liu, 2014*; *Hornung et al., 2002*) out of all the PBMC constituent cell types, this likely reflects an association of high baseline monocyte *TLR4* expression with risk. This finding was unexpected, as we have previously hypothesized that increasing TLR expression and/or signaling may help augment RTS,S/AS01 VE (*Moncunill et al., 2020*).

We have also previously reported that interferon, NF-κB, TLR, and monocyte-related BTMs were associated with protection in children and infants in the RTS,S/AS01 phase 3 trial (*Andersen-Nissen et al., 2021*). While the latter appears to contradict the identification of monocyte-related BTMs in the baseline risk signature identified in the present study, key differences between the two studies can account for this apparent discrepancy. The main difference is that in the present study we identified a baseline signature in vehicle-treated PBMC that associated directly with malaria risk, whereas we found hardly any associations when analyzing antigen-specific transcriptional responses. In contrast, in our previous study (*Moncunill et al., 2020*), we only assessed in vitro recall responses with antigen stimulation of samples obtained at 1 month after the third vaccination, correcting for background responses for each individual. Thus, the association of expression of monocyte-related BTMs with protection was observed after analyzing gene expression levels in CSP-stimulated, background-corrected PBMC. In the case of the vaccine-nonspecific antigen AMA1, we also observed a general pattern of inverse correlations with risk in vehicle-treated vs. AMA-1 stimulated PBMC (*Figure 3*). Another potential reason for why no BTMs were found to associate with the response to RTS,S/AS01

vaccination or with protection when analyzing CSP-stimulated PBMC is that all PBMC were stimulated on site for 12 hr (this stimulation time was chosen based on the kinetics of the IFN-γ transcriptional response) and then cryopreserved. Thus, we were unable to detect earlier transient responses that had already resolved by 12 hr, as well as more delayed response that had not yet initiated by 12 hr, if such responses occurred.

It is perhaps counterintuitive – considering that the RTS,S/AS01 vaccine does not contain AMA1 – that we observed a small number of BTMs associated with the response to RTS,S/AS01 vaccination and with clinical malaria risk when analyzing AMA1-stimulated PBMC. To explain this result, we refer the reader to our previous work that showed that RTS,S/AS01 vaccination alters antibody responses to antigens not contained in the RTS,S/AS01 vaccine (*Dobaño et al., 2019b*). RTS,S/AS01 recipients received partial protection from the RTS,S/AS01 vaccine, leading possibly to decreased *P. falciparum* parasite load and/or exposure (infection). We hypothesize that the AMA1 stimulation activated T cells that had been previously primed by prior exposure to *P. falciparum* and that RTS,S/AS01 recipients had fewer primed T cells due to decreased *P. falciparum* infection (via partial RTS,S/AS01 protection), providing a potential explanation for the transcriptional differences in AMA1-stimulated PBMC between RTS,S/AS01 vs. comparator recipients.

Compared to the 45 BTMs whose baseline levels significantly associated with clinical malaria risk in RTS,S/AS01-vaccinated African children, fewer BTMs (seven) had levels at 1-month postfinal RTS,S/AS01 dose that significantly associated with clinical malaria risk. Moreover, if a more stringent FDR cutoff had been used (i.e., 5%), six of these seven BTMs would not have been identified. Thus, it is entirely possible that, at 1-month postfinal RTS,S/AS01 dose, there is no circulating immune transcriptomic signature predictive of risk. Such a conclusion would not be surprising, given that in malaria-naive adults, the transcriptional response to the third RTS,S/AS01 dose has been shown to peak at day 1 postinjection, with some decline by day 6 and approximately 90% of the response having waned by day 21 (*Kazmin et al., 2017*). Therefore, it is likely that the sampling scheme in this study (1-month postfinal dose) misses the majority of the transcriptional response to RTS,S/AS01. Future studies with dense PBMC sampling during the transcriptional peak of the vaccine-induced response could be useful for further investigating RTS,S transcriptional immune correlates.

Additional limitations of our study include the following: first, PBMCs were stimulated on site and then frozen. As each site performed the procedure separately, this renders our data susceptible to batch effects. However, a standardized SOP and shared reagents were used, decreasing the possibility of such effects. Moreover, an advantage of onsite stimulation of fresh PBMC is that it avoids the decrease in cell viability, and potential loss of detection of Ag-specific cells, that may have occurred if PBMC had been frozen, thawed, and then stimulated at a central location. Second, there was confounding between age and location. As all infants were from Manhiça and the majority of children were from Bagamoyo, it was not possible to examine the impact of age or clinical trial site on RTS,S/AS01 transcriptional response. Third, we do not know whether the controls were truly protected or whether they were never exposed to malaria in the first place. This limitation highlights the importance of our cross-study analysis, where all participants are exposed. Fourth, despite the relatively large size of the study, our statistical power was limited by the number of malaria cases with available samples; sampling additional controls would not have increased our statistical power. Fifth, as only patrolling cell subsets are present in PBMC, we were unable to detect potential signals from T cells, B cells, NK cells, and macrophages localized to an infection site including skin and liver or the immune memory compartment localized in secondary lymphoid organs. Finally, while it is not uncommon to use an FDR cutoff of 20% in high-dimensional immune correlates studies (e.g., *Andersen-Nissen et al., 2021*; *Liu et al., 2021*; *Lu et al., 2021*; *Haynes et al., 2012*; *Fletcher et al., 2016*; *Young et al., 2021*), our results should be interpreted with the requisite level of caution. However, we do note that many of our significant modules in the baseline risk analysis would have survived even lower FDR cutoffs (in many cases even a 1% cutoff), giving us a fair degree of confidence in our results. For example, of the seven monocyte-related BTMs whose baseline levels associated with risk, all would have survived a 5% FDR cutoff, and three even a 1% cutoff; likewise, of the four dendritic cell-related BTMs whose baseline levels associated with risk, all would have survived a 1% cutoff.

Despite these limitations, our study also has a number of strengths. For example, while excellent work has already been done to interrogate transcriptional responses to RTS,S/AS01 vaccination in healthy, malaria-naive adults (including densely sampled early postvaccination sampling timepoints

to capture innate responses) and to identify molecular correlates of RTS,S/AS01-mediated protection against clinical malaria after CHMI in malaria-naive adults (*Kazmin et al., 2017*; *Du et al., 2020*; *Vahey et al., 2010*; *van den Berg et al., 2017*), in our study we examined transcriptional responses to RTS,S/AS01 vaccination in infants and children in malaria-endemic areas. This feature is a strength of our study, as (1) infants in particular have relatively immature immune systems (*Simon et al., 2015*), making it likely that infants (and younger children) mount different vaccine responses than adults (*Pichichero, 2014*) (2) infants and children are especially susceptible to malaria-related morbidity and mortality, making them the target population for this and other malaria vaccines; and (3) continual exposure to *P. falciparum*, as occurs in endemic areas, influences naturally acquired immunity, which in turn interacts with immunity conferred by RTS,S vaccination (*Dobaño et al., 2019b*). Related to this, another advantage of our study is the use of a comparator group which allows to discern the effect of the vaccine from environmental exposures including *P. falciparum* and age. As participants in the study are very young, significant development of their immune systems occurs throughout the duration of the study, meaning that such changes could potentially be confounded with vaccine-induced immune changes.

While it will be necessary to perform follow-up studies at more sites and with larger sample sizes to validate the baseline transcriptional signature associated with malaria risk identified here, our study suggests that innate immune cells may shape responses to RTS,S/AS01 and raises hypotheses for future testing related to monocytes and RTS,S/AS01-mediated protection.

## Acknowledgements

We are very grateful to study participants, their families, and vaccine trial site field and lab staff. We thank the phase 3 trial sites PIs Salim Abdulla, Pedro Alonso, Jahit Sacarlal, and Pedro Aide; the investigators involved in the generation of immunology data used here, including providers of antigens for antibody assays (Itziar Ubillos, Marta Vidal, Alfons Jimenez, Ruth Aguilar, Diana Barrios, Laura Puyol, Aintzane Ayestaran, Luis Izquierdo, David Cavanagh, James Beeson, David Lanar, Vir Chauhan, Chetan Chitnis, Deepak Gaur, Evelina Angov, Benoit Gamain, and Ross Coppel); the MAL067 Vaccine Immunology Consortium investigators and Working Groups; and Fergal Duffy for valuable comments on the manuscript. We thank GlaxoSmithKline Biologicals SA for their support in the conduct of the MAL067 study.

## Additional information

### Competing interests

Joseph J Campo: is an employee of Antigen Discovery Inc. The author declare that no other competing interests exist. The other authors declare that no competing interests exist.

### Funding

| Funder | Grant reference number | Author |
| --- | --- | --- |
| National Institute of Allergy and Infectious Diseases | R01AI095789 | Carlota Dobaño |
| National Institute of Allergy and Infectious Diseases | U19AI128914 | Julie McElrath Raphael Gottardo Ken Stuart |
| PATH Malaria Vaccine Initiative | Research Contract | Carlota Dobaño |
| Instituto de Salud Carlos III | PI11/00423 | Carlota Dobaño |
| National Institute of Allergy and Infectious Diseases | U19AI110818 | Daniel E Neafsey |
| Departament de Salut, Generalitat de Catalunya | SLT006/17/00109 | Gemma Moncunill |
| Instituto de Salud Carlos III | PI14/01422 | Carlota Dobaño |

| Funder | Grant reference number | Author |
| --- | --- | --- |
| Bill and Melinda Gates Foundation | INV-008576/OPP1154739 | Raphael Gottardo |
| Ministerio de Economía, Industria y Competitividad, Gobierno de España | RYC-2008-02631 | Carlota Dobaño |
| Instituto de Salud Carlos III | CD010/00156 | Gemma Moncunill |

The funders had no role in study design, data collection, and interpretation, or the decision to submit the work for publication.

## Author contributions

Gemma Moncunill, Conceptualization, Funding acquisition, Investigation, Resources, Supervision, Visualization, Writing – original draft, Writing – review and editing, Methodology; Jason Carnes, Investigation, Methodology, Resources, Writing – review and editing; William Chad Young, Data curation, Formal analysis, Investigation, Project administration, Visualization, Writing – review and editing; Lindsay Carpp, Investigation, Visualization, Writing – original draft, Writing – review and editing, Conceptualization; Stephen De Rosa, Conceptualization, Investigation, Methodology, Resources, Supervision, Writing – review and editing; Joseph J Campo, Selidji T Agnandji, Conceptualization, Funding acquisition, Investigation, Resources, Writing – review and editing; Augusto Nhabomba, Maxmillian Mpina, Chenjerai Jairoce, Investigation, Resources, Writing – review and editing; Greg Finak, Data curation, Formal analysis, Visualization, Writing – review and editing; Paige Haas, Investigation, Writing – review and editing, Methodology; Carl Muriel, Data curation, Formal analysis, Resources, Visualization, Writing – review and editing; Phu Van, Formal analysis, Investigation, Resources, Visualization, Writing – review and editing; Héctor Sanz, Data curation, Formal analysis, Writing – review and editing; Sheetij Dutta, Project administration, Resources, Writing – review and editing; Benjamin Mordmüller, Conceptualization, Funding acquisition, Investigation, Project administration, Resources, Writing – review and editing; Núria Díez-Padrisa, Project administration, Writing – review and editing; Nana Aba Williams, Funding acquisition, Project administration, Writing – review and editing; John J Aponte, Conceptualization, Data curation, Formal analysis, Investigation, Resources, Supervision, Writing – review and editing; Clarissa Valim, Conceptualization, Data curation, Formal analysis, Supervision, Writing – review and editing; Daniel E Neafsey, Conceptualization, Data curation, Funding acquisition, Investigation, Methodology, Project administration, Resources, Supervision, Writing – review and editing; Claudia Daubenberger, Carlota Dobaño, Conceptualization, Funding acquisition, Investigation, Project administration, Resources, Supervision, Writing – review and editing; M Juliana McElrath, Conceptualization, Funding acquisition, Investigation, Supervision, Writing – review and editing, Project administration; Ken Stuart, Conceptualization, Funding acquisition, Investigation, Project administration, Supervision, Writing – review and editing; Raphael Gottardo, Conceptualization, Data curation, Formal analysis, Investigation, Methodology, Resources, Project administration, Supervision, Methodology, Writing – review and editing

## Author ORCIDs

Gemma Moncunill http://orcid.org/0000-0001-5105-9836
Lindsay Carpp http://orcid.org/0000-0003-0333-5925
Daniel E Neafsey http://orcid.org/0000-0002-1665-9323
Carlota Dobaño http://orcid.org/0000-0002-6751-4060
Ken Stuart http://orcid.org/0000-0002-4064-9758
Raphael Gottardo http://orcid.org/0000-0002-3867-0232

## Ethics

Clinical trial registration ClinicalTrials.gov identifier NCT00866619.
The study protocol was approved by the Ethical Committee of the Hospital Clínic in Barcelona (CEIC, Spain), the National Health and Bioethics Committee (CNBS, Mozambique), the Ethikkommission Beider Basel (EKBB, Switzerland), the National Institutional Review Board (NIMR, Tanzania), the Ifakara Health Institute IRB (IHIIRB, Tanzania), the Lambaréné Independent Regional Ethics Committee (CERIL, Gabon), and the Research Ethics Committee (REC, USA). The study teams complied with the

Declaration of Helsinki and Good Clinical Practice including monitoring of data. Written informed consent was obtained from children's parents or guardians before recruitment.

### Decision letter and Author response
Decision letter https://doi.org/10.7554/eLife.70393.sa1
Author response https://doi.org/10.7554/eLife.70393.sa2

---

## Additional files

### Supplementary files
• Supplementary file 1. Complete information on study participant site, case–control matching ID, age cohort, sex assigned at birth, vaccine group, date of first dose vaccination, and case–control status.

• Supplementary file 2. Numbers, age group, and case–control status of RTS,S/AS01 recipients by site for whom months 0 and/or 3 peripheral blood mononuclear cell (PBMC) samples were included in the ICS/immunophenotyping analysis.

• Supplementary file 3. List of the 68 'RTS,S/AS01 signature BTMs' (Comparison 1 in *Figure 2*) tested as immune correlates.

• Supplementary file 4. List of 35 individual genes whose baseline expression in vehicle-treated peripheral blood mononuclear cell (PBMC) significantly associated with risk in MAL067. These individual genes were obtained by looking at gene-level correlate heatmaps (*Figure 6—figure supplements 2–9*) of the eight blood transcriptional modules (BTMs) (M4.0, S10, S5, M168, M4.3, M11.0, M4.15, and S4) that significantly associated with risk in MAL067 and in two controlled human malaria infection (CHMI studies). The table also contains information on whether each individual gene also significantly associated with risk in a CHMI study (check mark; columns 2 through 4).

• Transparent reporting form

### Data availability
Sequencing data have been deposited in GEO under accession code GSE176156. Immune phenotyping and intracellular cytokine staining data used for the analysis are archived on ImmPort (https://immport.niaid.nih.gov/home). The analysis code can be found at https://github.com/william-c-young/mal067_paper (copy archived at swh:1:rev:1168525a46d42019c8ba39d1c440e92e2c33c596).

The following dataset was generated:

| Author(s) | Year | Dataset title | Dataset URL | Database and Identifier |
|---|---|---|---|---|
| Moncunill G, Carnes J, Young WC, Carpp LN, Rosa SD, Campo JJ, Nhabomba AJ, Mpina M, Jairoce C, Finak G, Haas P, Murie C, Van P, Sanz H, Dutta S, Mordmüller B, Agnandji ST, Díez-Padrisa N, Williams NA, Aponte JJ, Valim C, Neafsey DE, Daubenberger C, McElrath J, Dobaño C, Stuart K, Gottardo R | 2021 | A baseline transcriptional signature associates with clinical malaria risk in RTS,S/AS01-vaccinated African children | https://www.ncbi.nlm.nih.gov/geo/query/acc.cgi?acc=GSE176156 | NCBI Gene Expression Omnibus, GSE176156 |

The following previously published datasets were used:

| Author(s) | Year | Dataset title | Dataset URL | Database and Identifier |
|---|---|---|---|---|
| Vahey M | 2009 | Expression data from a malaria vaccine trial (HG-U133A2.0 and U133 Plus 2.0) | https://www.ncbi.nlm.nih.gov/geo/query/acc.cgi?acc=GSE18323 | NCBI Gene Expression Omnibus, GSE18323 |
| Kazmn D, Pulendran B | 2017 | Systems analysis of protective immune responses to RTS,S malaria vaccination in humans | https://www.ncbi.nlm.nih.gov/geo/query/acc.cgi?acc=GSE89292 | NCBI Gene Expression Omnibus, GSE89292 |
| Du Y, Zak DE, Shankar S | 2020 | Transcriptional responses to RTS,S standard regimen (RRR) and Ad35.CS+R,R vaccination in controlled human malaria infection study | https://www.ncbi.nlm.nih.gov/geo/query/acc.cgi?acc=GSE103401 | NCBI Gene Expression Omnibus, GSE103401 |
| Thompson E, Du Y, Zak DE, Shankar S | 2020 | Gene responses to RTS,S malaria vaccination in controlled human malaria study | https://www.ncbi.nlm.nih.gov/geo/query/acc.cgi?acc=GSE102288 | NCBI Gene Expression Omnibus, GSE102288 |
| Du Y, Thompson E, Zak DE, Shankar S | 2020 | Transcriptional responses to RTS,S induced vaccination in controlled human malaria infection studies | https://www.ncbi.nlm.nih.gov/geo/query/acc.cgi?acc=GSE107672 | NCBI Gene Expression Omnibus, GSE107672 |

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
