## [Decision Letter]

**Decision letter after peer review:**

Thank you for submitting your article "A baseline transcriptional signature associates with clinical malaria risk in RTS,S/AS01-vaccinated African children" for consideration by *eLife*. Your article has been reviewed by 2 peer reviewers, and the evaluation has been overseen by a Reviewing Editor and Betty Diamond as the Senior Editor. The following individual involved in review of your submission has agreed to reveal their identity: Wiebke Nahrendorf (Reviewer #1).

*Reviewer #1 (Recommendations for the authors):*

The authors use a systems vaccinology approach to find a circulating immune signature that can predict RTS,S vaccine efficacy. It is perfectly possible that such a signature simply does not exist (the null hypothesis). I would recommend throughout the paper for the authors to check if their results (in the way they were generated and analysed) support or reject the null hypothesis – both of which are equally valid outcomes!

The MAL067 trial set up is fantastic (age-matched children receiving RTS,S or comparator vaccine who do or do not develop clinical malaria), but unfortunately isolating PBMCs 1-month after final vaccination to assess recall responses did not really work: whilst unstimulated PBMCs are transcriptional different between RTS,S and comparator, no signifiant signature of RTS,S vaccination after CSP and HBS stimulation was observed (Figure 2A). It is biologically hard to explain why differences were not maintained after stimulation. Together with the inconsistencies (opposite correlations) when analysing AMA-1 stimulated PBMCs or matched pre- and post RTS,S vaccination samples (Figure 2B), and allowing for 20% false discoveries it raises questions about the experimental and statistical robustness of this approach. Since antigen-stimulation of PBMCs is not informative the authors could focus their analysis on unstimulated PBMCs from Figure 3 onwards, which would massively streamline their message.

Figure 4: include Figure S2 as a panels as they convey crucial information: explicitly draw the conclusion that the production of multiple cytokines after 1-month after final RTS,S vaccination does not predict protection from clinical malaria (reflect in relevant results subheading and figure legend title).

Figure 5: this Figure is too dense which makes it hard to find the interesting information – it could be simplified by only including unstimulated PBMCs. It is surprising that antibodies do not correlate with B cell modules – causing some doubt about this correlation approach (not clear from line 229 – 232 if Spearman or Pearson? very relaxed p-value of 0.2).

Figure 6: A the five post vaccination modules identified in Figure 3 do not associate with risk in RTS,S CHMI studies – why might this be? young African children in malaria endemic environment vs malaria-naive US adults? or because these modules were not very strongly associated with risk in the first place? B – this panel is what the title and main conclusions of the paper are based on – it is interesting to speculate that the innate responses before vaccination may shape vaccine efficacy. Drilling down into the genes in the modules (functional transcriptomics) would be useful to get a glimpse into how this might work and to guide future work where e.g monocytes could be directly isolated. As it stands the conclusion that "inflammatory monocytes may inhibit protective RTS,S/AS01-induced responses" (line 58/59) is not well supported – especially since the authors published a paper last year which finds an inverse association (Moncunill G, Scholzen A, Mpina M, Nhabomba A, Hounkpatin AB, Osaba L, Valls R, Campo JJ, Sanz H, Jairoce C, et al: Antigen-stimulated PBMC transcriptional protective signatures for malaria immunization. Sci Transl Med 2020). The explanations offered in the discussion (621 – 633) are technical rather than biological again raising some questions about the robustness of the here presented approach.

General remarks

The decision to use blood transcriptional module analysis to reduce data dimensionality is a good one, but once candidate modules are identified the authors should drill down into the nitty gritty of which genes are up and downregulated and infer cell function to provide testible hypothesis for future functional and mechanistic studies.

Use figure legend headings/ results subheadings to summarise results.

Include n in Figure legends.

"month 3" could be called "post vaccination" for a more intuitive read.

Unstimulated PBMCs: sometimes called vehicle, sometimes DMSO – standardise.

Specifics

Line 319 "tended to correlate in opposite directions with risk" : no talk about malaria risk yet.

Line 466 – 475 move trial details from Results section to methods.

Supplementary methods are 3 lines long – include in text.

*Reviewer #3 (Recommendations for the authors):*

I thank the authors for presenting and interesting and well-written manuscript. In addition to my comments above, I have the following suggestions which I believe would improve the clarity and transparency of the paper

1. It would be helpful to provide an explanation of how the sample sizes were derived. Since the main objective was to identify BTMs associated with developing malaria after RTS,S/AS01 vaccine, it would be useful to report the malaria incidence in the main Phase 3 trial for the age groups studied at each site in the follow-up period of this study in the RTS,S and comparator vaccine arms. This will help to address the question of whether receipt of RTS,S vaccine is the main reason for the difference between cases and controls, or whether it is more related to exposure to infectious bites. If the EIR is low, then one might expect that the vaccine explains rather little of the difference between cases and controls, and therefore a large sample size would be needed to detect any significant association between gene expression and vaccine efficacy.

2. It would be helpful to include more data in table 1 to show how well the matching process for cases and controls worked. Table 1 could be reformatted to be easier to read, with separate columns for each site (rather than the current presentation of this information in parentheses) and could include rows for age, sex, time of vaccination and duration of follow-up.

3. In view of the perplexing result mentioned in the previous review section for DMSO stimulation, I think further detail about the PBMC stimulation is needed. I cannot find the DMSO concentration used for vehicle or antigen stimulation in either this paper or the "previously described" reference (24). The authors do not describe that they have subtracted or adjusted for the vehicle-induced gene expression in assessing antigen induced gene expression, but this might be one explanation for this odd result. If this is not the explanation, I worry that different concentrations of DMSO may have been used in the vehicle and antigen stimulation conditions. As this results appears anomalous, it requires further explanation of the methods or discussion as to why this result may have occurred.

4. The choice of an FDR significance threshold of 0.2 has not been justified. As mentioned in the previous section of the review this is extremely liberal, and diminishes confidence in the significance of the findings. In most analyses there are BTMs with FDRs below 0.05, so I think it is essential that the authors explain why they chose to use an FDR of 0.2 throughout the paper, and that they discuss the implications of selecting such a liberal threshold.

5. The authors have described quite a lot of different statistical approaches in the methods (line 227-256) but it is currently difficult for the reader to understand where each of these has been applied to the data that are presented (particularly in the figures). I think it is essential that figure legends include the number of subjects from each group included in analyses and the type of analysis which has been performed and the criteria for statistical significance. There are several elements in the current description of these analyses which did not fully make sense or which I could not see used in the results that have been presented:

a. Line 229-231 state that Spearman's rank correlation was used and then in the next sentence that Pearson correlation was used – please clarify as only one or the other should be used for each analysis.

b. Line 235-239 states that adjustment was made for clinical and experimental covariates – these covariates do not appear to be reported anywhere and it is unlcear in the Results section which analyses (if any) included this adjustment.

c. Line 247-250: the results of this logistic regression are not obviously presented (or at least this method of analysis is not reported explicitly in the Results section) and the stratification variables do not appear to have been reported anywhere.

6. The authors have decided to include combine rabies and meningococcal C vaccine arms into a single "comparator" vaccine group. One might expect these to elicit quite different effects on the immune system and I think it is important to present data to justify the decision to include them together. For example, a series of PCA plots of the DMSO-stimulated transcriptional responses, with subjects coloured by vaccine, by age group, and by site could be very informative for interpretation of the rest of the data in the paper.

7. In all figures the use of a log10 FDR colour scale in the heatmaps makes it very difficult to identify the FDR values. I appreciate that these are included in the supplementary tables, but is it possible to add the FDR value in each cell of the heatmap for the significant values, or if not, perhaps to adopt a categorical approach to colouring them (eg. FDR<0.01; FDR<0.05; FDR <0.2)? This would enhance interpretation.

---

## [Author Response]

Reviewer #1 (Recommendations for the authors):The authors use a systems vaccinology approach to find a circulating immune signature that can predict RTS,S vaccine efficacy. It is perfectly possible that such a signature simply does not exist (the null hypothesis). I would recommend throughout the paper for the authors to check if their results (in the way they were generated and analysed) support or reject the null hypothesis – both of which are equally valid outcomes!The MAL067 trial set up is fantastic (age-matched children receiving RTS,S or comparator vaccine who do or do not develop clinical malaria), but unfortunately isolating PBMCs 1-month after final vaccination to assess recall responses did not really work: whilst unstimulated PBMCs are transcriptional different between RTS,S and comparator, no signifiant signature of RTS,S vaccination after CSP and HBS stimulation was observed (Figure 2A). It is biologically hard to explain why differences were not maintained after stimulation.

The fact that bulk transcriptional profiling of Ag-stimulated PBMCs (and specifically to CSP) did not identify large significant differences in BTM expression between the RTS,S vs. comparator group could be due to several factors. First of all, the frequency of antigen-specific CD4^+^ T cells was very low among CD4^+^ T cells (Figure 4 of the manuscript shows that CSP-specific CD4^+^ T cells comprise < 0.004% of all CD4^+^ T cells). This low frequency of CSP-specific T cells is consistent with other RTS,S studies [e.g. as we state on line 385, we have previously found that CSP-specific T-cells in RTS,S/AS01 vaccinees comprise < 0.10% of all CD4^+^ T cells (1)]. Moreover, CD4^+^ T cells themselves comprise approximately 45-57% of all PBMCs (2). Thus, finding an expression signal between the RTS,S vs. comparator group would require the signal to be high enough to be detected in only 0.002% of all PBMCs [0.004% (% CSP-specific CD4^+^ T cells out of total CD4^+^ T cells) x 51% (average % of CD4^+^ T cells out of all PBMCs) = 0.002%]. Thus, lack of detectable recall response does not mean lack of recall response. Moreover, as suggested below, we opted not to focus the rest of the manuscript on the Ag-stimulation results.

Second of all, the PBMCs were stimulated on site for 12h and then cryopreserved. This stimulation time was chosen based on the kinetics of IFN-g and IL-2 mRNA response (3), but other responses may have had different kinetics and thus have already resolved or have not yet occurred by the 12-h cryopreservation. We have added text in the manuscript to discuss these caveats (“Another potential reason for why no BTMs were found to associate with the response to RTS,S/AS01 vaccination or with protection when analyzing CSP-stimulated PBMC is that all PBMC were stimulated on site for 12 hours (this stimulation time was chosen based on the kinetics of the IFN-γ transcriptional response) and then cryopreserved. Thus, we were unable to detect earlier transient responses that had already resolved by 12 hours, as well as more delayed response that had not yet initiated by 12 hours, if such responses occurred.; lines 599-605).

It should be noted that in all our analyses, the stimulated results were adjusted for DMSO to focus on the antigen-specific response only. This would explain why we detect signal in the DMSO samples but not in response to stimulation. We have realized that this was not very well described in the figure captions and the Methods section and have added more details, including the model description in Methods section. As such, we do not believe that these results impact all downstream conclusions. We believe that the unstimulated results provide significant new insights into the immune and molecular mechanisms of RTS,S vaccine efficacy, not necessarily directly related to the RTS,S-specific acquired immune response. Finally, we would like to highlight the fact that we have improved our model specification to directly account for the pairing of some of the samples using a random effect using the limma package. This has slightly increased statistical power, and as such the number of significantly differentially expressed BTMs in response to stimulation is a bit higher (but still much less than that for the DMSO). Originally, we had decided against the use of a random effect due to the computational cost of estimating the random effect.

In addition, one clarification, we investigated whether a circulating immune signature associates with individual-level clinical malaria case/control status, not RTS,S vaccine efficacy.

Together with the inconsistencies (opposite correlations) when analysing AMA-1 stimulated PBMCs or matched pre- and post RTS,S vaccination samples (Figure 2B), and allowing for 20% false discoveries it raises questions about the experimental and statistical robustness of this approach.

Our response explains the opposite correlations. Since AMA1 is actually looking at the effect of AMA1-DMSO.

As for the FDR rate, it is not uncommon to use a threshold of 20% for immune correlates studies [e.g. (6-11)]. We agree with you that it is important to clearly state the chosen FDR rate and to discuss conclusions in the context of the FDR rate used. We see we could improve our manuscript in this respect. We have added the following:

Results: “Compared to the 45 BTMs whose baseline levels significantly associated with clinical malaria risk in RTS,S/AS01-vaccinated African children, fewer BTMs (seven) had levels at one month post-final RTS,S/AS01 dose that significantly associated with clinical malaria risk. Moreover, if a more stringent FDR cutoff had been used (i.e. 5%), six of these seven BTMs would not have been identified. Thus it is entirely possible that, at one month post-final RTS,S/AS01 dose, there is no circulating immune transcriptomic signature predictive of risk…” (lines 617-628)

Discussion: “Finally, while it is not uncommon to use an FDR cutoff of 20% in high-dimensional immune correlates studies [e.g. (65-70)], our results should be interpreted with the requisite level of caution. However, we do note that many of our significant modules in the baseline risk analysis would have survived even lower FDR cutoffs (in many cases even a 1% cutoff), giving us a fair degree of confidence in our results. For example, of the seven monocyte-related BTMs whose baseline levels associated with risk, all would have survived a 5% FDR cut-off, and three even a 1% cut-off; likewise, of the four dendritic cell-related BTMs whose baseline levels associated with risk, all would have survived a 5% FDR cut-off, and three even a 1% cut-off.” (lines 644-651)

Moreover, we have revised Figures 2, 3, and 6 so that it is easy to discern whether a specific BTM correlation would also pass more stringent FDR cutoffs, through the addition of 1, 2, or 3 asterisks where appropriate: “|FDR| < 0.2 (*), < 0.05 (**), < 0.01 (***).” Note that, most central to the key message of the paper, many of the monocyte-related, DC-related, and cell cycle-related BTMs would have passed more stringent FDR cutoffs, with many even passing a 1% FDR cutoff (as discussed above).

Since antigen-stimulation of PBMCs is not informative the authors could focus their analysis on unstimulated PBMCs from Figure 3 onwards, which would massively streamline their message.

We believe that the stimulation results should be reported in Figure 3 but agree that the manuscript could be streamlined by removing subsequent antigen stimulation results. We have made the following revisions to the manuscript:

– Removed all data from Ag-stimulated PBMC from Figure 5

– Removed related discussion of Ag-stimulation results

– Removed all data from Ag-stimulated PBMC from Figure S2

– Removed all data from Ag-stimulated PBMC from Figure S11

Figure 4: include Figure S2 as a panels as they convey crucial information: explicitly draw the conclusion that the production of multiple cytokines after 1-month after final RTS,S vaccination does not predict protection from clinical malaria (reflect in relevant results subheading and figure legend title).

We have modified Figure 4 as suggested and made the requested edits to the results subheading and figure title.

Figure 5: this Figure is too dense which makes it hard to find the interesting information – it could be simplified by only including unstimulated PBMCs. It is surprising that antibodies do not correlate with B cell modules – causing some doubt about this correlation approach (not clear from line 229 – 232 if Spearman or Pearson? very relaxed p-value of 0.2).

Thank you for this good suggestion. We have modified Figure 5 as requested.

The fact that expression levels of B-cell-related modules in PBMC circulating at month 3 do not correlate with antibodies circulating at the same time is not surprising given that antibodies are secreted by antigen-specific B cells that constitute a very small percentage of total B cells (and thus an even smaller percentage of PBMC).

Moreover, the samples were collected 1 month post vaccination, which is past the plasma cell response. Typically, plasmablast response and B cell activation occurs about one week post-vaccination [e.g. (12-15)]; the effect on circulating antibodies is seen later, e.g. Nakaya et al. showed that, in influenza vaccine recipients, the frequency of influenza-specific IgG secreting plasmablasts at day 7 correlates significantly with the influenza antibody response at day 28 (12). In the RTS,S study in naïve adults (4), activation was detected after 1 and 2 days after the administration of RTS,S vaccine doses. In that study, antigen-specific plasmablasts were detected after 6 days after vaccination but frequencies of these cells did not correlate with the antibody response.

Figure 6: A the five post vaccination modules identified in Figure 3 do not associate with risk in RTS,S CHMI studies – why might this be? young African children in malaria endemic environment vs malaria-naive US adults? or because these modules were not very strongly associated with risk in the first place?

It is true that in panel A there is no clear pattern for the whole 7 modules across all the RTS,S CHMI studies (see discussion below and the manuscript text for potential explanations of this). In the text related to Figure 6A we state:

“No BTM was consistently associated with malaria risk (or non-protection) across all four studies. The most consistent result was for the monocyte-related BTM “enriched in monocytes (II) (M11.0),” whose month 3 expression was significantly associated with risk in two of the three CHMI studies (Figure 6A, Figure 6-source data 1).” (lines 480-483).

As to the differences between study populations and whether this may explain the lack of a clear pattern for the whole 7 modules in Figure 6A across all the CHMI studies, the study populations clearly differ with respect to a variety of factors including but not limited to age, genetics, malaria exposure/malaria history, and other infection history. Thus, yes, we believe that these differences are one of the main reasons for the observed differences between studies. In addition, samples from the Phase 3 trial were collected a month after the 3^rd^ dose, whereas samples in the other studies were collected at different times (21 or 14 days post-vaccination), which could result also in some differences between studies. Finally, there were differences in the formulation of the RTS,S vaccine (AS01E adjuvant vs AS02A or AS01B adjuvants). All these differences are already explained in the manuscript.

As to the question of whether these seven modules were “not very strongly associated with risk in the first place”, this is also possible. As discussed above, only one of these seven modules would have passed a more stringent FDR cutoff of 5%. We now state in the Discussion:

“Compared to the 45 BTMs whose baseline levels significantly associated with clinical malaria risk in RTS,S/AS01-vaccinated African children, fewer BTMs (seven) had levels at one month post-final RTS,S/AS01 dose that significantly associated with clinical malaria risk. Moreover, if a more stringent FDR cutoff had been used (i.e. 5%), six of these seven BTMs would not have been identified. Thus it is entirely possible that, at one month post-final RTS,S/AS01 dose, there is no circulating immune transcriptomic signature predictive of risk. Such a conclusion would not be surprising, given that in malaria-naïve adults, the transcriptional response to the third RTS,S/AS01 dose has been shown to peak at Day 1 post-injection, with some decline by Day 6 and approximately 90% of the response having waned by Day 21 (17). Therefore, it is likely that the sampling scheme in this study (one month post-final dose) misses the majority of the transcriptional response to RTS,S/AS01. Future studies with dense PBMC sampling during the transcriptional peak of the vaccine-induced response could be useful for further investigating RTS,S transcriptional immune correlates.” (lines 617-628)

B – this panel is what the title and main conclusions of the paper are based on – it is interesting to speculate that the innate responses before vaccination may shape vaccine efficacy. Drilling down into the genes in the modules (functional transcriptomics) would be useful to get a glimpse into how this might work and to guide future work where e.g monocytes could be directly isolated. As it stands the conclusion that "inflammatory monocytes may inhibit protective RTS,S/AS01-induced responses" (line 58/59) is not well supported – especially since the authors published a paper last year which finds an inverse association (Moncunill G, Scholzen A, Mpina M, Nhabomba A, Hounkpatin AB, Osaba L, Valls R, Campo JJ, Sanz H, Jairoce C, et al: Antigen-stimulated PBMC transcriptional protective signatures for malaria immunization. Sci Transl Med 2020). The explanations offered in the discussion (621 – 633) are technical rather than biological again raising some questions about the robustness of the here presented approach.

Thank you for this excellent suggestion. We have discussed above the gene-level analyses we have done and the new supplementary figures and supplementary file that have been added to the manuscript.

Please note that the hypothesis that inflammatory monocytes specifically inhibit RTS,S vaccine efficacy was originally put forth by Warimwe et al. in 2013 “It is plausible that RTS,S vaccine efficacy is specifically inhibited by inflammatory monocytes, thus confounding induction of an effective adaptive response, but further studies in both animal models and humans will be needed to confirm this.” (16) We mention this in our Discussion:

“Based on previous evidence in various mouse models of viral infection that inflammatory monocytes inhibit T cell proliferation (54, 55), T cell activation (55), and B cell responses (56), Warimwe et al. proposed that inflammatory monocytes may inhibit RTS,S-induced protective adaptive responses.” (lines 564-567).

Prior to performing the gene-level analyses, the fact that we did observe upregulation of many monocyte-related BTMs at baseline in cases vs controls (Figure 6B), as well as many inflammatory-related BTMs at baseline in cases vs controls (Figure 6B), did seem to support this hypothesis.

We agree that our previous hypothesis that “inflammatory monocytes may inhibit protective RTS,S/AS01-induced responses” should be revised in light of our stabilin-1 (clever-1) findings, discussed above. In fact, if the increased expression of STAB1 at baseline in PBMC in cases vs controls (Figure 6—figure supplement 1, 2, and 6) is reflective of an increase in circulating stabilin-1^high^ immunosuppressive monocytes (this would of course need to be confirmed experimentally), this would support an opposite hypothesis that immunosuppressive monocytes may inhibit protective RTS,S/AS01-induced responses.

We have made the following edits to remove the mentions of “inflammatory monocytes”:

Abstract (lines 71-72): “suggesting that inflammatory monocytes may inhibit protective RTS,S/AS01-induced responses” γ edited to “suggests that certain monocyte subsets may inhibit protective RTS,S/AS01-induced responses”.

We have also removed the discussion of Mitchell et al. showing a potential association of inflammatory monocytes and decreased vaccine immunity, as well as removing the suggestion of potentially modulating monocyte populations via chemokine receptor antagonists to help boost RTS,S efficacy*.*

We have added the following to the Discussion (lines 567-578):

“Our gene-level correlates analyses suggest an alternative hypothesis, however. With the caveat that the gene-level analyses were performed post hoc, high baseline expression of *STAB1* (which is present in DC-related, monocyte-related, and cell cycle-related modules) was found to positively associate with clinical malaria risk. *STAB1* encodes stabilin-1 (also called Clever-1), a transmembrane glycoprotein scavenger receptor that links extracellular signals to intracellular vesicle trafficking pathways (17). Interestingly, stabilin-1^high^ monocytes show downregulation of proinflammatory genes, and T cells co-cultured with stabilin-1^high^ monocytes showed decreased antigen recall, suggesting that monocyte stabilin-1 suppresses T cell activation (18). Thus one possibility is that stabilin-1^high^ immunosuppressive monocytes circulating at baseline could decrease protective RTS,S-induced T-cell responses, or inhibit another aspect of adaptive immunity. Single-cell transcriptomic profiling of PBMC in future RTS,S trials in African children in malaria-endemic areas could help test this hypothesis.”

Please note that we stress that this is a “possibility” and a “hypothesis” rather than a definitive conclusion of the work; as we state at the beginning of the Discussion:

“Our main finding is the identification of a baseline blood transcriptional module (BTM) signature that associates with clinical malaria risk in RTS,S/AS01-vaccinated African children. In a cross-study comparison, much of this baseline risk signature – specifically, dendritic cell- and monocyte-related BTMs – was also recapitulated in two of the three CHMI studies in healthy, malaria-naïve adults.” (lines 543-546)

This conclusion is well-supported by our data.

Regarding how the results of the present manuscript relate to the results of Moncunill et al. 2020 STM: We see how the two sets of results may seem discrepant; however, when we examine the specific details of our previous and current analyses, we see that they are in fact compatible.

Author response table 1 summarizes the technical differences and similarities between the relevant work described in the present manuscript and the work described in Moncunill et al. 2020 STM.

**Author response table 1. sa2table1:** 

	Present manuscript: Baseline signature associated with risk	Moncunill et al. 2020 STM: Protective signature
PBMC sampling timepoint	Baseline	1 month post-third vaccination
PBMC stimulation	Vehicle (DMSO); stimulated on site before cryopreservation of cell pellets for subsequent RNA extraction.	24-hour antigen (CSP) stimulation adjusted by vehicle (DMSO); PBMC were cryopreserved before stimulation
Background correction	N/A	Yes, subtraction of expression in vehicle-stimulated PBMC (thus the observed response is specific to antigen stimulation, i.e. recall response)
Presence of monocyte-related BTMs	Yes, of the 45 BTMs, 7 were monocyte-related: M81, M118.1, M11.0, M118.0, S4, M73, M4.15	Yes, of the 24 BTMs, 3 were monocyte-related: M81, M118.1, M11.0
Gene expression measurement	RNA-seq	Microarray
Hypothesis:	Stabilin-1^high^ immunosuppressive monocytes circulating at baseline may inhibit protective RTS,S-induced T-cell responses [supported by (18)] or another RTS,S-induced protective adaptive response.	Protected individuals may have monocytes that are qualitatively superior in mediating, e.g., Fc receptor/antibody-dependent responses (reflected by an altered transcriptional profile), resulting in improved control of infection.

We have added the following text to our Discussion (lines 584-605):

“We have also previously reported that interferon, NF-κB, TLR, and monocyte-related BTMs were associated with protection in children and infants in the RTS,S/AS01 phase 3 trial (64). […] Thus, we were unable to detect earlier transient responses that had already resolved by 12 hours, as well as more delayed response that had not yet initiated by 12 hours, if such responses occurred.”

General remarksThe decision to use blood transcriptional module analysis to reduce data dimensionality is a good one, but once candidate modules are identified the authors should drill down into the nitty gritty of which genes are up and downregulated and infer cell function to provide testible hypothesis for future functional and mechanistic studies.

We have added new supplementary figures (Figure 6—figure supplements 1-8) looking at individual genes. These analyses did yield interesting results, especially with respect to STAB1 (stabilin-1, clever-1). We have significantly revised our Discussion in the light of these new results.

Use figure legend headings/ results subheadings to summarise results

Currently, our results subheadings are written in conclusion form e.g. “RTS,S/AS01 vaccination is associated with month 3 downregulation of B cell- and monocyte-related BTMs, along with upregulation of T cell-related BTMs”, “Monocyte-related RTS,S/AS01 signature BTMs associate with clinical malaria risk”, “RTS,S/AS01 vaccination elicits polyfunctional CSP-specific CD4^+^ T-cell responses elicited that do not correlate with malaria risk”, “Month 3 levels of RTS,S/AS01 signature BTMs tend to correlate directly with month 3 IgM antibody responses and inversely with month 3 IgG responses”, and “Cross-study immune correlates analysis reveals a mostly consistent association in RTS,S/AS01-vaccinees between baseline expression of DC- and monocyte-related BTMs and risk”.

We prefer to keep our figure titles descriptive for two reasons: (1) avoiding redundancy with the results subheadings, and (2) we think it is most straightforward to describe what analysis is shown in the figure, and leave the interpretation/result from that figure to the “Results” text.

Include n in Figure legends.

We have included the “n”s in the legends of Figures 2, 3, 4, 5 and 6.

Reviewer #3 (Recommendations for the authors):I thank the authors for presenting and interesting and well-written manuscript. In addition to my comments above, I have the following suggestions which I believe would improve the clarity and transparency of the paper1. It would be helpful to provide an explanation of how the sample sizes were derived. Since the main objective was to identify BTMs associated with developing malaria after RTS,S/AS01 vaccine, it would be useful to report the malaria incidence in the main Phase 3 trial for the age groups studied at each site in the follow-up period of this study in the RTS,S and comparator vaccine arms. This will help to address the question of whether receipt of RTS,S vaccine is the main reason for the difference between cases and controls, or whether it is more related to exposure to infectious bites. If the EIR is low, then one might expect that the vaccine explains rather little of the difference between cases and controls, and therefore a large sample size would be needed to detect any significant association between gene expression and vaccine efficacy.

Malaria transmission intensity of each site in 2007 (2 years before the start of the study) and the malaria incidence during the Phase 3 Clinical Trial can be found in the final Phase 3 Clinical trial paper (21). Malaria transmission intensity in Bagamoyo and Manhiça sites was low/moderate. Therefore, we used a case-control design for the study instead of a cohort design. Sample sizes were based on availability of samples and malaria cases. We used all samples available from malaria cases and selected 2 to 4 matched controls for each case for RTS,S vaccinees and 2 controls for comparators. Higher numbers of controls were used to consider the heterogeneity and the lack of specificity of the controls as we do not know if these were truly protected children or they had not been exposed. Adding more controls would not increase statistical power.

We have made the following revisions to the text in this light:

Materials and methods: “As malaria transmission intensity at the Bagamoyo and Manhiça sites was low/moderate (11), we used a case-control design for the study instead of a cohort design. Sample sizes were based on availability of samples and malaria cases. We used all samples available from malaria cases and selected 2 to 4 matched controls for each case for RTS,S vaccinees and 2 controls for comparators. Higher numbers of controls were used to consider the heterogeneity and the lack of specificity of the controls as we do not know if these were truly protected children (they may have been sick but not gone to a health post, or they may not have been exposed during follow-up). In selecting controls, we prioritized participants who had samples at both month 0 and month 3 and in whom the complete set of antigen stimulations was conducted.” (lines 166-170)

Discussion (of limitations): “Third, we do not know whether the controls were truly protected or whether they were never exposed to malaria in the first place. This limitation highlights the importance of our cross-study analysis, where all participants are known to be exposed*.*” (lines 637-641)

2. It would be helpful to include more data in table 1 to show how well the matching process for cases and controls worked. Table 1 could be reformatted to be easier to read, with separate columns for each site (rather than the current presentation of this information in parentheses) and could include rows for age, sex, time of vaccination and duration of follow-up.

Thank you for the good suggestion to reformat Table 1. We have redone Table 1 to include separate columns for each site and agree that readability has improved substantially. In parallel, we have also reformatted Supplementary File 2 for improved readability.

Moreover, we now also provide the new Supplementary File 1, which provides complete information on participant match ID, site, age cohort, sex assigned at birth, and time of vaccination.

3. In view of the perplexing result mentioned in the previous review section for DMSO stimulation, I think further detail about the PBMC stimulation is needed. I cannot find the DMSO concentration used for vehicle or antigen stimulation in either this paper or the "previously described" reference (24). The authors do not describe that they have subtracted or adjusted for the vehicle-induced gene expression in assessing antigen induced gene expression, but this might be one explanation for this odd result. If this is not the explanation, I worry that different concentrations of DMSO may have been used in the vehicle and antigen stimulation conditions. As this results appears anomalous, it requires further explanation of the methods or discussion as to why this result may have occurred.

DMSO (D2650, Σ) was used at a final dilution of 1/322, the same concentration of DMSO as used for CSP peptide pool. We have now added this information to the manuscript. (lines 184-1485) As discussed above, we have also added details on how the stimulation data were analyzed (i.e. correcting for DMSO).

4. The choice of an FDR significance threshold of 0.2 has not been justified. As mentioned in the previous section of the review this is extremely liberal, and diminishes confidence in the significance of the findings. In most analyses there are BTMs with FDRs below 0.05, so I think it is essential that the authors explain why they chose to use an FDR of 0.2 throughout the paper, and that they discuss the implications of selecting such a liberal threshold.

While it is not uncommon to use a threshold of 20% for immune correlates studies [e.g. (6-11)], we agree with you that it is important to clearly state the chosen FDR rate and to discuss conclusions in the context of the FDR rate used. We see we could improve our manuscript in this respect. We have added the following:

Results: “Compared to the 45 BTMs whose baseline levels significantly associated with clinical malaria risk in RTS,S/AS01-vaccinated African children, fewer BTMs (seven) had levels at one month post-final RTS,S/AS01 dose that significantly associated with clinical malaria risk. Moreover, if a more stringent FDR cutoff had been used (i.e. 5%), six of these seven BTMs would not have been identified. Thus it is entirely possible that, at one month post-final RTS,S/AS01 dose, there is no circulating immune transcriptomic signature predictive of risk…” (lines 617-628)

Discussion: “Finally, while it is not uncommon to use an FDR cutoff of 20% in high-dimensional immune correlates studies [e.g. (65-70)], our results should be interpreted with the requisite level of caution. However, we do note that many of our significant modules in the baseline risk analysis would have survived even lower FDR cutoffs (in many cases even a 1% cutoff), giving us a fair degree of confidence in our results. For example, of the seven monocyte-related BTMs whose baseline levels associated with risk, all would have survived a 5% FDR cut-off, and three even a 1% cut-off; likewise, of the four dendritic cell-related BTMs whose baseline levels associated with risk, all would have survived a 5% FDR cut-off, and three even a 1% cut-off.” (lines 644-651)

Moreover, we have revised Figures 2, 3, and 6 so that it is easy to discern whether a specific BTM correlation would also pass more stringent FDR cutoffs, through the addition of 1, 2, or 3 asterisks where appropriate: “|FDR| < 0.2 (*), < 0.05 (**), < 0.01 (***).” Note that, most central to the key message of the paper, many of the monocyte-related, DC-related, and cell cycle-related BTMs would have passed more stringent FDR cutoffs, with many even passing a 1% FDR cutoff (as discussed above).

5. The authors have described quite a lot of different statistical approaches in the methods (line 227-256) but it is currently difficult for the reader to understand where each of these has been applied to the data that are presented (particularly in the figures). I think it is essential that figure legends include the number of subjects from each group included in analyses and the type of analysis which has been performed and the criteria for statistical significance. There are several elements in the current description of these analyses which did not fully make sense or which I could not see used in the results that have been presented:a. Line 229-231 state that Spearman's rank correlation was used and then in the next sentence that Pearson correlation was used – please clarify as only one or the other should be used for each analysis.b. Line 235-239 states that adjustment was made for clinical and experimental covariates – these covariates do not appear to be reported anywhere and it is unlcear in the Results section which analyses (if any) included this adjustment.c. Line 247-250: the results of this logistic regression are not obviously presented (or at least this method of analysis is not reported explicitly in the Results section) and the stratification variables do not appear to have been reported anywhere.

We have significantly improved our method description both in the results and methods section, as detailed below:

Figure legends: Numbers of participants included in each analysis have been included. Methods: Full equations for all analyses have been provided (“plain English” versions of these are now in the figure legends). Criteria for statistical significance are already present in the figure legends.

a) “Pearson” changed to “Spearman”

b) Edited to “All analyses controlled for plate, total reads, and age.” In addition, the full equations have been provided in Methods.

c) We have removed the results of this analysis and the related Methods text. Instead, for testing for significant differences in controls vs cases for each of the monocyte-related variables in Figure 6—figure supplement 10, we have added the following text to the legend of Figure 6—figure supplement 10:

“The p values at the bottom of each panel are from testing for a significant difference in controls vs cases within each panel, and were modeled using a mixed-effects model (using lmer) with match id as a random effect.”

Each panel also contains a p value underneath for the significance of the difference in cases vs controls.

6. The authors have decided to include combine rabies and meningococcal C vaccine arms into a single "comparator" vaccine group. One might expect these to elicit quite different effects on the immune system and I think it is important to present data to justify the decision to include them together. For example, a series of PCA plots of the DMSO-stimulated transcriptional responses, with subjects coloured by vaccine, by age group, and by site could be very informative for interpretation of the rest of the data in the paper.

While PCA plots could be generated as the reviewer suggests, there is a complete overlap of vaccine in comparators and age cohort (all infants received meningococcal vaccine and all children received rabies vaccine). Thus, we are not sure how informative such plots would be as they would be confounded by age group.

The impact of combining both vaccines in comparator recipients on the study results and conclusions is minimal since the main results are based on baseline gene expression and its association with malaria risk within RTS,S vaccinees. Correlates of malaria risk in comparators are done separately. Comparator vaccination may be a confounding factor for age cohort, but we are not analyzing the effect of age cohort on the transcriptional profile. Comparators are only included in the analysis of RTS,S immunogenicity at post-vaccination (RTS,S vs Comparators, Figure 2A, Comparison (1)) and we have adjusted analyses by age cohort and hence by comparator vaccine. The fact that the comparators received different control vaccines only stresses that the BTMs found to be associated with RTS,S vaccination are specific to the RTS,S vaccine.

Moreover, as an alternative way to identify RTS,S-specific transcriptional responses, we also include Comparison (2), which compares Month 3 to Month 0 transcription levels within RTS,S vaccinees.

7. In all figures the use of a log10 FDR colour scale in the heatmaps makes it very difficult to identify the FDR values. I appreciate that these are included in the supplementary tables, but is it possible to add the FDR value in each cell of the heatmap for the significant values, or if not, perhaps to adopt a categorical approach to colouring them (eg. FDR<0.01; FDR<0.05; FDR <0.2)? This would enhance interpretation.

Yes, Figures 2, 3, 3—figure supplement 1, and 6 now have asterisks to denote three different FDR cutoffs: |FDR| < 0.2 (*), < 0.05 (**), < 0.01 (***).

References:

1. Moncunill G, De Rosa SC, Ayestaran A, Nhabomba AJ, Mpina M, Cohen KW, Jairoce C, Rutishauser T, Campo JJ, Harezlak J, Sanz H, Diez-Padrisa N, Williams NA, Morris D, Aponte JJ, Valim C, Daubenberger C, Dobano C, McElrath MJ. RTS,S/AS01E Malaria Vaccine Induces Memory and Polyfunctional T Cell Responses in a Pediatric African Phase III Trial. Front Immunol. 2017;8:1008.

2. Kleiveland CR. Peripheral Blood Mononuclear Cells. In: Verhoeckx K, Cotter P, López-Expósito I, Kleiveland C, Lea T, Mackie A, et al., editors. The Impact of Food Bioactives on Health: in vitro and ex vivo models. Cham: Springer International Publishing; 2015. p. 161-7.

3. Schultz-Thater E, Frey DM, Margelli D, Raafat N, Feder-Mengus C, Spagnoli GC, Zajac P. Whole blood assessment of antigen specific cellular immune response by real time quantitative PCR: a versatile monitoring and discovery tool. J Transl Med. 2008;6:58.

4. Kazmin D, Nakaya HI, Lee EK, Johnson MJ, van der Most R, van den Berg RA, Ballou WR, Jongert E, Wille-Reece U, Ockenhouse C, Aderem A, Zak DE, Sadoff J, Hendriks J, Wrammert J, Ahmed R, Pulendran B. Systems analysis of protective immune responses to RTS,S malaria vaccination in humans. Proc Natl Acad Sci U S A. 2017;114(9):2425-30.

5. Subramanian A, Tamayo P, Mootha VK, Mukherjee S, Ebert BL, Gillette MA, Paulovich A, Pomeroy SL, Golub TR, Lander ES, Mesirov JP. Gene set enrichment analysis: a knowledge-based approach for interpreting genome-wide expression profiles. Proc Natl Acad Sci U S A. 2005;102(43):15545-50.

6. Liu C, Martins AJ, Lau WW, Rachmaninoff N, Chen J, Imberti L, Mostaghimi D, Fink DL, Burbelo PD, Dobbs K, Delmonte OM, Bansal N, Failla L, Sottini A, Quiros-Roldan E, Han KL, Sellers BA, Cheung F, Sparks R, Chun TW, Moir S, Lionakis MS, Consortium NC, Clinicians C, Rossi C, Su HC, Kuhns DB, Cohen JI, Notarangelo LD, Tsang JS. Time-resolved systems immunology reveals a late juncture linked to fatal COVID-19. Cell. 2021;184(7):1836-57 e22.

7. Andersen-Nissen E, Fiore-Gartland A, Ballweber Fleming L, Carpp LN, Naidoo AF, Harper MS, Voillet V, Grunenberg N, Laher F, Innes C, Bekker LG, Kublin JG, Huang Y, Ferrari G, Tomaras GD, Gray G, Gilbert PB, McElrath MJ. Innate immune signatures to a partially-efficacious HIV vaccine predict correlates of HIV-1 infection risk. PLoS Pathog. 2021;17(3):e1009363.

8. Lu P, Guerin DJ, Lin S, Chaudhury S, Ackerman ME, Bolton DL, Wallqvist A. Immunoprofiling Correlates of Protection Against SHIV Infection in Adjuvanted HIV-1 Pox-Protein Vaccinated Rhesus Macaques. Front Immunol. 2021;12:625030.

9. Haynes BF, Gilbert PB, McElrath MJ, Zolla-Pazner S, Tomaras GD, Alam SM, Evans DT, Montefiori DC, Karnasuta C, Sutthent R, Liao HX, DeVico AL, Lewis GK, Williams C, Pinter A, Fong Y, Janes H, DeCamp A, Huang Y, Rao M, Billings E, Karasavvas N, Robb ML, Ngauy V, de Souza MS, Paris R, Ferrari G, Bailer RT, Soderberg KA, Andrews C, Berman PW, Frahm N, De Rosa SC, Alpert MD, Yates NL, Shen X, Koup RA, Pitisuttithum P, Kaewkungwal J, Nitayaphan S, Rerks-Ngarm S, Michael NL, Kim JH. Immune-correlates analysis of an HIV-1 vaccine efficacy trial. N Engl J Med. 2012;366(14):1275-86.

10. Fletcher HA, Snowden MA, Landry B, Rida W, Satti I, Harris SA, Matsumiya M, Tanner R, O'Shea MK, Dheenadhayalan V, Bogardus L, Stockdale L, Marsay L, Chomka A, Harrington-Kandt R, Manjaly-Thomas ZR, Naranbhai V, Stylianou E, Darboe F, Penn-Nicholson A, Nemes E, Hatherill M, Hussey G, Mahomed H, Tameris M, McClain JB, Evans TG, Hanekom WA, Scriba TJ, McShane H. T-cell activation is an immune correlate of risk in BCG vaccinated infants. Nat Commun. 2016;7:11290.

11. Young WC, Carpp LN, Chaudhury S, Regules JA, Bergmann-Leitner ES, Ockenhouse C, Wille-Reece U, deCamp AC, Hughes E, Mahoney C, Pallikkuth S, Pahwa S, Dennison SM, Mudrak SV, Alam SM, Seaton KE, Spreng RL, Fallon J, Michell A, Ulloa-Montoya F, Coccia M, Jongert E, Alter G, Tomaras GD, Gottardo R. Comprehensive Data Integration Approach to Assess Immune Responses and Correlates of RTS,S/AS01-Mediated Protection From Malaria Infection in Controlled Human Malaria Infection Trials. Front Big Data. 2021;4:672460.

12. Nakaya HI, Wrammert J, Lee EK, Racioppi L, Marie-Kunze S, Haining WN, Means AR, Kasturi SP, Khan N, Li GM, McCausland M, Kanchan V, Kokko KE, Li S, Elbein R, Mehta AK, Aderem A, Subbarao K, Ahmed R, Pulendran B. Systems biology of vaccination for seasonal influenza in humans. Nat Immunol. 2011;12(8):786-95.

13. Frolich D, Giesecke C, Mei HE, Reiter K, Daridon C, Lipsky PE, Dorner T. Secondary immunization generates clonally related antigen-specific plasma cells and memory B cells. J Immunol. 2010;185(5):3103-10.

14. Li S, Sullivan NL, Rouphael N, Yu T, Banton S, Maddur MS, McCausland M, Chiu C, Canniff J, Dubey S, Liu K, Tran V, Hagan T, Duraisingham S, Wieland A, Mehta AK, Whitaker JA, Subramaniam S, Jones DP, Sette A, Vora K, Weinberg A, Mulligan MJ, Nakaya HI, Levin M, Ahmed R, Pulendran B. Metabolic Phenotypes of Response to Vaccination in Humans. Cell. 2017;169(5):862-77 e17.

15. Wrammert J, Smith K, Miller J, Langley WA, Kokko K, Larsen C, Zheng NY, Mays I, Garman L, Helms C, James J, Air GM, Capra JD, Ahmed R, Wilson PC. Rapid cloning of high-affinity human monoclonal antibodies against influenza virus. Nature. 2008;453(7195):667-71.

16. Warimwe GM, Fletcher HA, Olotu A, Agnandji ST, Hill AV, Marsh K, Bejon P. Peripheral blood monocyte-to-lymphocyte ratio at study enrollment predicts efficacy of the RTS,S malaria vaccine: analysis of pooled phase II clinical trial data. BMC Med. 2013;11:184.

17. Kzhyshkowska J, Gratchev A, Goerdt S. Stabilin-1, a homeostatic scavenger receptor with multiple functions. J Cell Mol Med. 2006;10(3):635-49.

18. Palani S, Elima K, Ekholm E, Jalkanen S, Salmi M. Monocyte Stabilin-1 Suppresses the Activation of Th1 Lymphocytes. J Immunol. 2016;196(1):115-23.

19. Moncunill G, Scholzen A, Mpina M, Nhabomba A, Hounkpatin AB, Osaba L, Valls R, Campo JJ, Sanz H, Jairoce C, Williams NA, Pasini EM, Arteta D, Maynou J, Palacios L, Duran-Frigola M, Aponte JJ, Kocken CHM, Agnandji ST, Mas JM, Mordmuller B, Daubenberger C, Sauerwein R, Dobano C. Antigen-stimulated PBMC transcriptional protective signatures for malaria immunization. Sci Transl Med. 2020;12(543).

20. Moodie Z, Juraska M, Huang Y, Zhuang Y, Fong Y, Carpp LN, Self SG, Chambonneau L, Small R, Jackson N, Noriega F, Gilbert PB. Neutralizing Antibody Correlates Analysis of Tetravalent Dengue Vaccine Efficacy Trials in Asia and Latin America. J Infect Dis. 2018;217(5):742-53.

21. RTS, S Clinical Trials Partnership. Efficacy and safety of RTS,S/AS01 malaria vaccine with or without a booster dose in infants and children in Africa: final results of a phase 3, individually randomised, controlled trial. Lancet. 2015;386(9988):31-45.